# ToRA: A Tool-Integrated Reasoning Agent for Mathematical Problem Solving

**Zhibin Gou**[1,2*], **Zhihong Shao**[1,2*], **Yeyun Gong**[2†], **Yelong Shen**[3]
**Yujiu Yang**[1†], **Minlie Huang**[1†], **Nan Duan**[2], **Weizhu Chen**[3]
[1]Tsinghua University [2]Microsoft Research [3]Microsoft Azure AI
{gzb22,szh19}@mails.tsinghua.edu.cn
{yegong,yeshe,nanduan,wzchen}@microsoft.com

## Abstract

Large language models have made significant progress in various language tasks, yet they still struggle with complex mathematics. In this paper, we propose ToRA, a series of Tool-integrated Reasoning Agents designed to solve challenging mathematical problems by seamlessly integrating natural language reasoning with the utilization of external tools (e.g., computation libraries and symbolic solvers), thereby amalgamating the analytical prowess of language and the computational efficiency of tools. To train ToRA, we curate interactive tool-use trajectories on mathematical datasets, apply imitation learning on the annotations, and propose output space shaping to further refine models' reasoning behavior. As a result, ToRA models significantly outperform open-source models on 10 mathematical reasoning datasets across all scales with 13%-19% absolute improvements on average. Notably, ToRA-7B reaches 44.6% on the competition-level dataset MATH, surpassing the best open-source model WizardMath-70B by 22% absolute. ToRA-Code-34B is also the first open-source model that achieves an accuracy exceeding 50% on MATH, which significantly outperforms GPT-4's CoT result, and is competitive with GPT-4 solving problems with programs. Additionally, we conduct a comprehensive analysis of the benefits and remaining challenges of tool interaction for mathematical reasoning, providing valuable insights for future research[1].

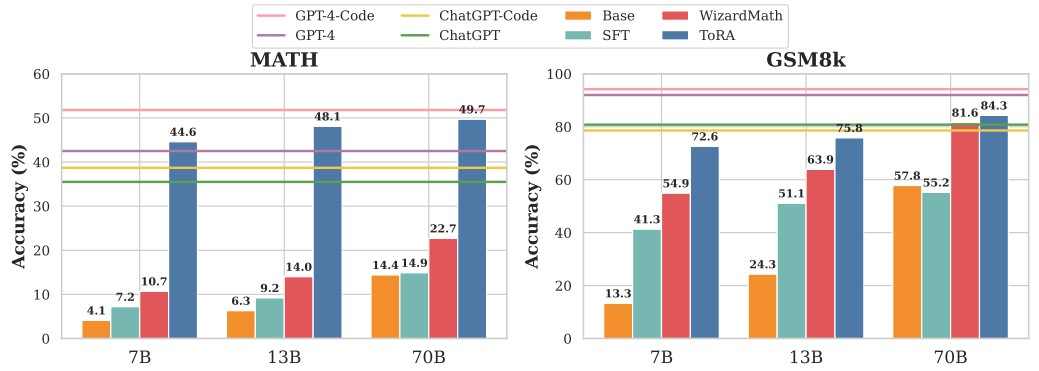

Figure 1: Comparing ToRA with baselines on LLaMA-2 base models from 7B to 70B. ToRA models exhibit remarkable improvements over previous state-of-the-art approaches across all scales. In particular, ToRA-70B notably outperforms GPT-4's CoT result on MATH and attains comparable results to GPT-4 solving problems with code.

---

[*]Equal contribution. See Contributions section for details. Work done during an internship at Microsoft.
[†]Corresponding authors.
[1]Code and models are available at https://github.com/microsoft/ToRA.

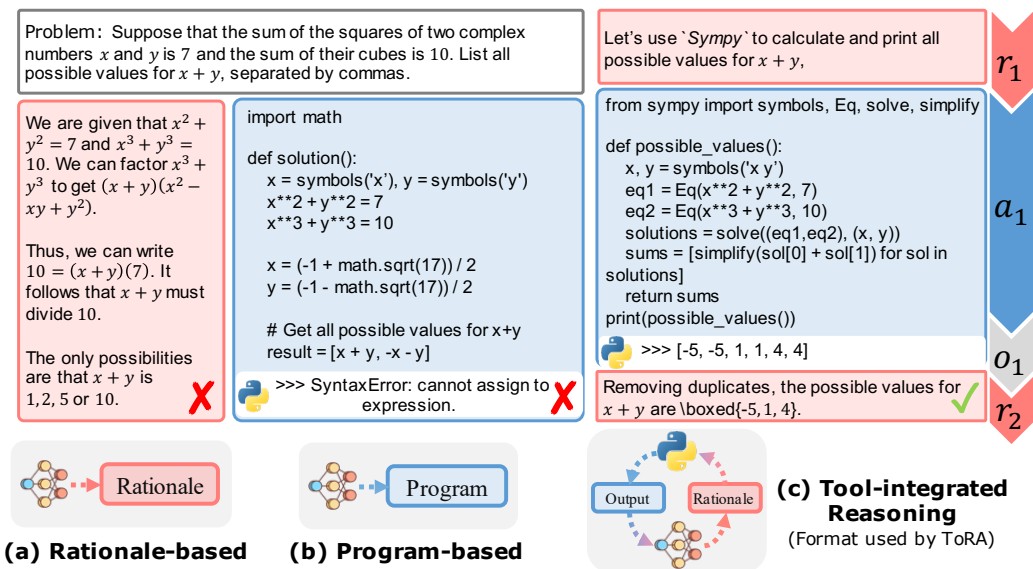

Figure 2: Examples of three reasoning formats for mathematical reasoning: (a) Rationale-based methods (e.g., CoT prompting) generate step-by-step natural language rationales, (b) Program-based methods (e.g., PAL prompting) solve tasks with program synthesis, and (c) our proposed Tool-integrated Reasoning format interleaves rationales with program-based tool use. For brevity, we present a simple example of *single-round tool interaction*, where the model creates rationale $r_1$ for analysis, writes program $a_1$ to call an external solver, obtains the execution output $o_1$, and then generates rationale $r_2$ to finalize the answer.

# 1 INTRODUCTION

Large language models (LLMs), such as GPT-4 (OpenAI, 2023) and PaLM-2 (Anil et al., 2023), have demonstrated remarkable progress in a wide range of language tasks, particularly in the longstanding challenge of mathematical reasoning (Feigenbaum et al., 1963; Hosseini et al., 2014). However, open-source models, such as LLaMA-2 (Touvron et al., 2023a;b) and Falcon (Penedo et al., 2023), still struggle with advanced mathematical reasoning tasks.

Existing works improve mathematical performance of language models either with step-by-step natural language reasoning (Wei et al., 2022) as illustrated in Fig 2 (a), or by synthesizing and executing programs to obtain the answers (Gao et al., 2022; Chen et al., 2022), as depicted in Fig 2 (b). Both approaches exhibit complementary advantages. Natural language is suitable for semantic analysis, planning, and abstract reasoning (e.g., commonsense reasoning), but struggles with precise computation, symbolic manipulation, and algorithmic processing. Conversely, programs excel in rigorous operations, and can outsource intricate calculations to specialized tools like equation solvers.

To leverage the benefits of both natural language reasoning and program-based tool use, we train open-source models such as LLaMA-2 to reason in a way where natural language reasoning is interleaved with program-based tool use synergistically (as depicted in Fig 2 (c)), thereby largely reducing the gap with closed-source models like GPT-4 in mathematical reasoning. Specifically, we first design the interleaving format of reasoning, curate corresponding interactive tool-use trajectories for mathematical problems from the popular GSM8k (Cobbe et al., 2021) and MATH (Hendrycks et al., 2021) dataset, and then apply imitation learning on the high-quality annotations, leading to a better performance than any existing open-source model. Furthermore, since the curated data is far from exhausting all valid trajectories for a problem, relying solely on imitation learning restricts a model's output space, hindering the flexibility in exploring plausible trajectories during testing. To improve the diversity of plausible reasoning steps and mitigate improper tool-use behavior, we apply *output space shaping* which additionally trains the models on both self-sampled valid trajectories and invalid ones that have been corrected by a teacher model (e.g., a 34B model can serve as the teacher

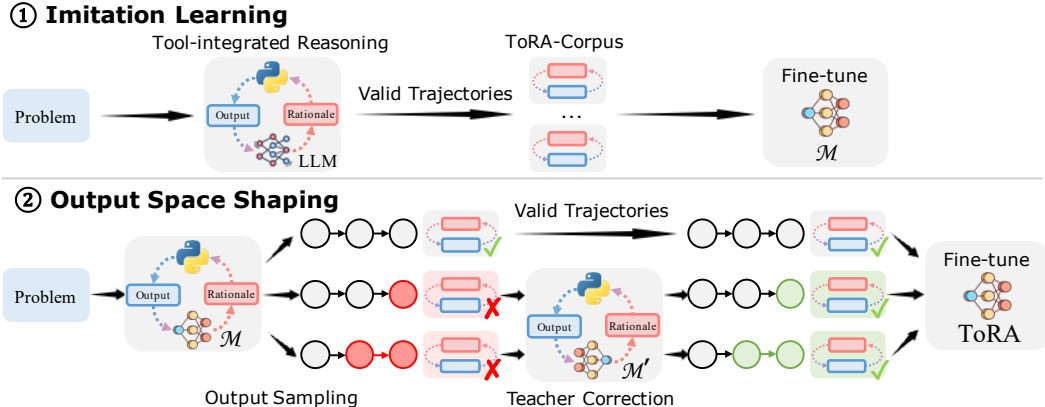

Figure 3: Training TORA contains two steps. ① **Imitation Learning**: Prompt LLMs like GPT-4 to generate Tool-integrated Reasoning trajectories (TORA-CORPUS) and use this corpus to fine-tune a model $\mathcal{M}$; ② **Output Space Shaping**: Sample diverse tool-use trajectories with $\mathcal{M}$, keep the valid ones, correct the invalid ones with a teacher model $\mathcal{M}'$, and retrain $\mathcal{M}$ on the union of sampled valid trajectories, corrected ones, and the initial TORA-CORPUS to obtain TORA.

for a 7B model). *Output space shaping* significantly boosts reasoning, allowing open-source models to attain an accuracy exceeding 50% on the competition-level MATH dataset for the first time.

We evaluate the resulting suite of Tool-integrated Reasoning Agents (TORA) ranging from 7B to 70B on 10 diverse mathematical reasoning datasets. As shown in Fig 1, TORA series significantly outperform open-source models across all scales. Notably, on the competition-level MATH dataset, TORA-7B outperforms the previous SoTA WizardMath-70B (Luo et al., 2023) by 22% absolute. TORA-CODE-34B beats GPT-4's CoT result (Bubeck et al., 2023) by 8.3% absolute (50.8% vs. 42.5%), and is competitive with GPT-4 solving problems with code (GPT-4-Code, 51.8%). In addition, we analyze the benefits and remaining challenges of tool interaction for mathematical reasoning, providing valuable insights for future work.

## 2 TORA: TOOL-INTEGRATED AGENTS FOR MATHEMATICAL REASONING

### 2.1 OVERVIEW

TORA series solve challenging mathematical problems by leveraging both natural language reasoning and program-based tool use. As shown in Fig 2 (c), given a mathematical problem $q$, TORA reasons with natural language, producing $r_1$. When reaching a point where program-based tool use is more appropriate for the subsequent task, e.g., equation solving, TORA generates a program $a_1$ for tool use following natural language guidance $r_1$. The execution output $o_1$ will be fed to TORA for subsequent processing including tool use adjustments, sub-tasks solving, or answer finalization. We repeat the process until the model places its answer within "`\boxed{}`". The resulting trajectory is denoted as $\tau = r_1 a_1 o_1 ... r_{n-1} a_{n-1} o_{n-1} r_n$, where $r_n$ contains the answer.

Fig 3 presents the training pipeline of TORA. We first collect interactive tool-use trajectories on popular mathematical datasets. We then apply imitation learning on the resulting annotations, as well as output space shaping to further refine models' reasoning behavior.

### 2.2 COLLECTING INTERACTIVE TOOL-USE TRAJECTORIES

Existing mathematical reasoning datasets primarily contain annotations in either natural language or code, posing a challenge for training tool-integrated agents due to the absence of interactive tool-use annotations. To address this, we utilize GPT-4 to synthesize high-quality trajectories on the GSM8k and MATH training sets. We select GSM8k and MATH as they exhibit diverse reasoning patterns, spanning multiple domains and difficulty levels.

---
**Algorithm 1** Inference of Tool-Integrated Reasoning

---
**Require:** problem $q$, model $\mathcal{G}$, prompt $p$, external tools $\mathcal{E}$, stop condition $Stop(\cdot)$, maximum iteration rounds $n$

1:   $\tau_0 \leftarrow$ ""                                                       ▷ Trajectory Initialization
2:   **for** $i \leftarrow 1$ to $n$ **do**
3:       $r_i \sim \mathbb{P}_{\mathcal{G}}(\cdot | p \oplus q \oplus \tau_{i-1})$                               ▷ Rationale Generation (Eq. 1)
4:       **if** $Stop(r_i)$ **then**                                               ▷ Stopping Criteria
5:          **return** $\tau_{i-1} \oplus r_i$
6:       **end if**
7:       $a_i \sim \mathbb{P}_{\mathcal{G}}(\cdot | p \oplus q \oplus \tau_{i-1} \oplus r_i)$                   ▷ Program Generation (Eq. 2)
8:       $o_i \leftarrow \mathcal{E}(a_i)$                                                  ▷ Tool Execution
9:       $\tau_i \leftarrow \tau_{i-1} \oplus r_i \oplus a_i \oplus o_i$                        ▷ Trajectory Update (Eq. 3)
10: **end for**
11: **return** $\tau_n$

---

**Prompt Curation**    We compose instructions along with diverse few-shot examples, utilizing an interleaved format as depicted in Fig 2 (c). These examples showcase interactive tool usage trajectories, incorporating descriptive variable names and combined program outputs. Please refer to Appendix E for the assembled prompts.

**Inference Procedure**    We follow Algorithm 1 and feed GPT-4 ($\mathcal{G}$) with the composed prompt $p$ to generate a tool-use trajectory $\tau$ for each question $q$ from the training set. The trajectory is initialized as an empty string $\tau_0$, for each interaction round $i$, we first generate a rationale:

$$r_i \sim \mathbb{P}_{\mathcal{G}}(\cdot | p \oplus q \oplus \tau_{i-1}) \tag{1}$$

where $\oplus$ means concatenation. If $r_i$ includes an answer within "\boxed{}" (i.e., the stopping condition $Stop(r_i)$), we cease generation, otherwise the model continues to write a program for tool use:

$$a_i \sim \mathbb{P}_{\mathcal{G}}(\cdot | p \oplus q \oplus \tau_{i-1} \oplus r_i) \tag{2}$$

In line with Gou et al. (2023), if the model triggers the code execution stop words like "```output", we supply it with the corresponding execution message and output $o_i$ by calling tools with $o_i \leftarrow \mathcal{E}(a_i)$, facilitating the generation of subsequent steps. Then, we update the trajectory by concatenating it with the newly generated rationale $r_i$, program $a_i$, and output $o_i$:

$$\tau_i \leftarrow \tau_{i-1} \oplus r_i \oplus a_i \oplus o_i \tag{3}$$

We repeat the above interaction process until we reach the maximum rounds $n$.

**Trajectory Sampling**    We set $n = 3$ and perform inference using GPT-4 with greedy decoding, retaining trajectories that yield correct answers. For questions where GPT-4 fails with greedy decoding, we apply nucleus sampling with a sample size of 10 and keep up to 4 valid trajectories per question. Ultimately, we successfully annotate trajectories for 98.2% of GSM8k questions and 83.1% of MATH questions. After filtering out invalid trajectories with tool-use errors or wrong answers, we obtain 16k annotations which constitute our dataset TORA-CORPUS. Table 1 compares TORA-CORPUS with recently proposed mathematical reasoning datasets, while Table 6 in the Appendix displays MATH annotation accuracy details.

## 2.3 TRAINING

**Imitation Learning**    We apply imitation learning on TORA-CORPUS by minimizing negative log-likelihood loss on the trajectory $\tau$ conditioned on the problem $q$:

$$\mathcal{M} = \arg\min_{\mathcal{M}} \sum_{q,\tau} \sum_{i=1}^{n-1} -\log \mathbb{P}_{\mathcal{M}}(r_{i+1} a_{i+1} | q, r_1...o_i) \tag{4}$$

where $\mathcal{M}$ is the resulting model. After imitation learning, we can simply apply the same procedure in Algorithm 1 by setting prompt to empty $p =$ "" for inference. Imitation learning leads to state-of-the-art mathematical reasoning performance despite the small scale of TORA-CORPUS.

Table 1: Compared with mathematical reasoning datasets, TORA-CORPUS uniquely combines natural language rationales with program-based tool usage. Note that TORA-CORPUS only employ questions from the original training set of MATH and GSM8k.

| Methods | #Annotation | Tool | Interleaving | LLM Used | Source |
|---|---|---|---|---|---|
| RFT (Yuan et al., 2023) | >100k | ✗ | ✗ | LLaMA-2 | GSM8k |
| Open-Platypus Lee et al. (2023) | 25k | ✗ | ✗ | GPT-4 | 11 datasets with MATH |
| WizardMath (Luo et al., 2023) | >96k | ✗ | ✗ | ChatGPT | MATH & GSM8k |
| Lila (Mishra et al., 2022) | 134k | ✓(PoT) | ✗ | - | 20 datasets with MATH & GSM8k |
| MathInstruct (Yue et al., 2023) | 260k | ✓(PoT) | ✗ | GPT-4 | 14 datasets with MATH & GSM8k |
| TORA-CORPUS (ours) | 16k | ✓ | ✓ | GPT-4 | MATH & GSM8k |

**Output Space Shaping**   For each question, TORA-CORPUS mostly demonstrates only one valid interactive tool-use trajectory, which may restrict a model's output space, rendering it inflexible in exploring plausible trajectories during testing. We therefore propose *output space shaping* in order to encourage the diversity of plausible reasoning steps and reduce improper tool-use behavior.

To explore diverse valid trajectories, we apply nucleus sampling to imitation learning models $\mathcal{M}$ to sample 64 trajectories per training question $q$, following the inference procedure in Section 2.2. We retain valid trajectories with correct answers and no tool-use errors. As many samples are duplicates, to further improve diversity and in an attempt to correct models' improper behavior, we seek to leverage invalid trajectories as well. We observe that trajectories with wrong answers are mostly incorrect halfway (Li et al., 2023), and the preceding reasoning is still plausible; in other words, we can obtain valid trajectories by correcting the subsequent portions. Specifically, a wrong trajectory $\widetilde{\tau}$, when written in text, can be represented as a sequence of lines separated by line breaks, i.e., $\widetilde{\tau} = l_1...l_m$, where $m$ is the total number of lines in $\widetilde{\tau}$. We enumerate possible preceding portions of wrong trajectories, i.e., $\widetilde{\tau}[: j] = l_1...l_j$, and leverage a teacher model $\mathcal{M}'$ to complete the subsequent steps with greedy decoding: $\tau \leftarrow \mathbb{P}_{\mathcal{M}'}(\cdot | q \oplus \widetilde{\tau}[: j])$ where we abuse the notation $\mathbb{P}_{\mathcal{M}'}(\cdot)$ to denote the interactive tool use process following Section 2.2. Finally, corrected trajectories as well as valid trajectory samples will be used for model training, thereby shaping the output space.

In our experiments, we always use CodeLLaMA-34B trained on TORA-CORPUS as the teacher model, and apply sampling with the CodeLLaMA series (ranging from 7B to 34B, with imitation learning). We obtain a total of 233k distinct valid trajectory samples and 69k corrected ones. From this combined dataset, we randomly select up to 4 trajectories per GSM8k and MATH problem, merge them with TORA-CORPUS, and then train all TORA models on the resulting 69k annotations.

## 3   EXPERIMENTS

### 3.1   IMPLEMENTATION DETAILS

We fine-tuned LLaMA-2 (Touvron et al., 2023b) and CodeLLaMA (Rozière et al., 2023) series (ranging from 7B to 70B) using TORA-CORPUS with output space shaping, yielding the TORA and TORA-CODE series respectively. We used a learning rate of 2e-5 by default except that we used 1e-5 for the 34B and 70B models. We set the global batch size to 128 and used a linear scheduler with a 3% warm-up period for 3 epochs. We trained all models with *DeepSpeed ZeRO Stage3* (Rajbhandari et al., 2021) and *Flash-Attention 2* (Dao, 2023). We used greedy decoding for all results, with the maximum sequence length set to 2,048 and the maximum number of tool executions set to 3.

### 3.2   EVALUATION SETUP

**Datasets** We evaluated models on GSM8k (Cobbe et al., 2021) and MATH (Hendrycks et al., 2021), along with 8 out-of-distribution datasets, namely GSM-Hard (Gao et al., 2022), SVAMP (Patel et al., 2021), ASDIV (Miao et al., 2020), TabMWP (Lu et al., 2023), SingleEQ, SingleOP, AddSub, and MultiArith (Koncel-Kedziorski et al., 2016), as illustrated in Table 5 in Appendix. The 10 assorted datasets collectively encompass mathematical problems spanning basic arithmetic to competition level, covering middle and high school curricula and various mathematical domains. The problem formats comprise tabular-based, free-form, and multiple-choice questions, ensuring a thorough assessment of the model's mathematical reasoning aptitude.

Table 2: Results on 10 mathematical reasoning tasks. MAWPS results are averaged over four tasks: Singleeq, Singleop, Addsub, and MultArith. Vanilla models are tested with CoT. The best results in each section are in blue, the second-best results are underlined, while the results of our best model are bolded. * ZS: Zero-shot inference without demonstrations.

| Model | Size | Tools | ZS* | GSM8k | MATH | GSM-Hard | SVAMP | TabMWP | ASDiv | MAWPS | AVG |
|---|---|---|---|---|---|---|---|---|---|---|---|
| **Used for training?** | | | | ✓ | ✓ | ✗ | ✗ | ✗ | ✗ | ✗ | |
| | | | | | | Proprietary Models | | | | | |
| GPT-4 | - | ✗ | ✗ | 92.0 | 42.5 | 64.7 | 93.1 | 67.1 | 91.3 | 97.6 | 78.3 |
| GPT-4 (PAL) 🐍 | - | ✓ | ✗ | 94.2 | 51.8 | 77.6 | 94.8 | 95.9 | 92.6 | 97.7 | 86.4 |
| ChatGPT | - | ✗ | ✗ | 80.8 | 35.5 | 55.9 | 83.0 | 69.1 | 87.3 | 94.6 | 72.3 |
| ChatGPT (PAL) 🐍 | - | ✓ | ✗ | 78.6 | 38.7 | 67.6 | 77.8 | 79.9 | 81.0 | 89.4 | 73.3 |
| Claude-2 | - | ✗ | ✗ | 85.2 | 32.5 | - | - | - | - | - | - |
| PaLM-2 | 540B | ✗ | ✗ | 80.7 | 34.3 | - | - | - | - | - | - |
| | | | | | | Open-Source Models | | | | | |
| LLaMA-2 | 7B | ✗ | ✗ | 13.3 | 4.1 | 7.8 | 38.0 | 31.1 | 50.7 | 60.9 | 29.4 |
| LLaMA-2 SFT | 7B | ✗ | ✓ | 41.3 | 7.2 | 16.1 | 31.9 | 27.8 | 47.4 | 60.0 | 33.1 |
| LLaMA-2 RFT | 7B | ✗ | ✓ | 51.2 | - | - | - | - | - | - | - |
| Platypus-2 | 7B | ✗ | ✗ | 14.4 | 5.4 | 8.6 | 36.7 | 26.5 | 47.9 | 58.4 | 28.3 |
| WizardMath | 7B | ✗ | ✓ | 54.9 | 10.7 | 20.6 | 57.3 | 38.1 | 59.1 | 73.7 | 44.9 |
| CodeLLaMA (PAL) 🐍 | 7B | ✓ | ✗ | 34.0 | 16.6 | 33.6 | 59.0 | 47.3 | 61.4 | 79.6 | 47.4 |
| Toolformer† 🧮 | 7B | ✓ | ✓ | - | - | - | 29.4 | - | 40.4 | 44.0 | - |
| ToRA 🐻 | 7B | ✓ | ✓ | 68.8 | 40.1 | 54.6 | 68.2 | 42.4 | 73.9 | 88.8 | 62.4 |
| ToRA-Code 🐻 | 7B | ✓ | ✓ | **72.6** | **44.6** | **56.0** | **70.4** | **51.6** | **78.7** | **91.3** | **66.5 (+19)** |
| LLaMA-2 | 13B | ✗ | ✗ | 24.3 | 6.3 | 13.6 | 43.1 | 39.5 | 56.3 | 70.4 | 36.2 |
| LLaMA-2 SFT | 13B | ✗ | ✓ | 51.1 | 9.2 | 22.3 | 46.3 | 35.8 | 58.6 | 75.0 | 42.6 |
| LLaMA-2 RFT | 13B | ✗ | ✓ | 55.3 | - | - | - | - | - | - | - |
| Platypus-2 | 13B | ✗ | ✗ | 23.7 | 7.1 | 14.3 | 50.7 | 45.3 | 55.1 | 69.6 | 38.0 |
| WizardMath | 13B | ✗ | ✓ | 63.9 | 14.0 | 28.4 | 64.3 | 46.7 | 65.8 | 79.7 | 51.8 |
| CodeLLaMA (PAL) 🐍 | 13B | ✓ | ✗ | 39.9 | 19.9 | 39.0 | 62.4 | 59.5 | 65.3 | 86.0 | 53.1 |
| ToRA 🐻 | 13B | ✓ | ✓ | 72.7 | 43.0 | 57.3 | 72.9 | 47.2 | 77.2 | 91.3 | 65.9 |
| ToRA-Code 🐻 | 13B | ✓ | ✓ | **75.8** | **48.1** | **60.5** | **75.7** | **65.4** | **81.4** | **92.5** | **71.3 (+18)** |
| LLaMA-1 RFT | 34B | ✗ | ✓ | 57.9 | - | - | - | - | - | - | - |
| CodeLLaMA (PAL) 🐍 | 34B | ✓ | ✗ | 53.3 | 23.9 | 49.4 | 71.0 | 63.1 | 72.4 | 91.5 | 60.7 |
| ToRA-Code 🐻 | 34B | ✓ | ✓ | **80.7** | **50.8** | **63.7** | **80.5** | **70.5** | **84.2** | **93.3** | **74.8 (+14)** |
| LLaMA-2 | 70B | ✗ | ✗ | 57.8 | 14.4 | 36.0 | 73.6 | 57.5 | 76.0 | 92.4 | 58.2 |
| LLaMA-2 SFT | 70B | ✗ | ✓ | 69.3 | 14.9 | 39.0 | 64.0 | 53.0 | 71.3 | 84.8 | 56.6 |
| LLaMA-2 RFT | 70B | ✗ | ✓ | 64.8 | - | - | - | - | - | - | - |
| Platypus-2 | 70B | ✗ | ✗ | 45.9 | 15.0 | 24.6 | 74.3 | 47.3 | 72.7 | 91.1 | 53.0 |
| WizardMath | 70B | ✗ | ✓ | 81.6 | 22.7 | 50.3 | 80.0 | 49.8 | 76.2 | 86.2 | 63.8 |
| LLaMA-2 (PAL) 🐍 | 70B | ✓ | ✗ | 55.2 | 18.3 | 50.0 | 74.6 | 59.5 | 71.9 | 92.8 | 60.3 |
| ToRA 🐻 | 70B | ✓ | ✓ | **84.3** | **49.7** | **67.2** | **82.7** | **74.0** | **86.8** | **93.8** | **76.9 (+13)** |

**Metrics** We report accuracies of predicted answers. Following Lightman et al. (2023), we round numerical values and use `sympy` [2] for parsing expressions. Since the SingleEQ, SingleOP, AddSub, and MultiArith datasets focus on different aspects of basic arithmetic, we report their average results under the collective term MAWPS (Koncel-Kedziorski et al., 2016) for all methods.

## 3.3 BASELINES

**Proprietary Models** We present results from an array of SoTA LLMs, such as OpenAI's GPT-4, ChatGPT (`gpt-3.5-turbo`), Google's PaLM-2, and Anthropic's Claude-2. By default, we report CoT prompting results, and include PAL (Gao et al., 2022) prompting results for selected models.

**Open-Source Models** *Base models* comprise LLaMA-2 and CodeLLaMA with CoT and PAL prompting. *Supervised Fine-Tuning (SFT)* employs CoT rationales from the original GSM8k and MATH dataset (15k samples) for fine-tuning. *Rejection sampling Fine-Tuning (RFT)* leverages multiple models to generate diverse reasoning paths for fine-tuning (Yuan et al., 2023). *WizardMath* augments data using ChatGPT, and conducts SFT and RLHF. *Platypus-2*, the top model on the LLM Leaderboard [3], is fine-tuned with Open-Platypus reasoning datasets (Lee et al., 2023). We also compare ToRA with Toolformer (Schick et al., 2023) which is a model trained to utilize calculators.

---

[2] https://www.sympy.org
[3] https://huggingface.co/spaces/HuggingFaceH4/open_llm_leaderboard

Table 3: Results on MATH subtopics.

| Model | Size | Tool | Intermediate Algebra | Precalculus | Geometry | Number Theory | Counting & Probability | Prealgebra | Algebra | Overall |
|---|---|---|---|---|---|---|---|---|---|---|
| | | | *Proprietary Models* | | | | | | | |
| ChatGPT (PAL) 🐍 | - | ✓ | 18.5 | 19.2 | 23.2 | 48.5 | 43.0 | 62.7 | 45.4 | 38.7 |
| GPT-4 (PAL) 🐍 | - | ✓ | 32.8 | 29.3 | 38.0 | 58.7 | 61.0 | 73.9 | 59.1 | 51.8 |
| | | | *Open-Source Models* | | | | | | | |
| WizardMath | 7B | ✗ | 6.2 | 6.0 | 6.5 | 7.6 | 9.5 | 18.1 | 16.3 | 11.2 |
| ToRA-Code 🐯 | 7B | ✓ | **35.1 (+28.9)** | **31.0 (+25.0)** | **24.0 (+17.5)** | **50.7 (+43.1)** | **30.6 (+21.1)** | **55.0 (+36.9)** | **61.7 (+45.4)** | **44.6 (+33.4)** |
| w/o Shaping | 7B | ✓ | 29.7 (-5.4) | 25.1 (-5.9) | 17.7 (-6.3) | 46.9 (-3.8) | 32.3 (+1.7) | 51.9 (-3.1) | 55.7 (-6.0) | 40.2 (-4.4) |
| w/o Rationale | 7B | ✓ | 25.5 (-9.6) | 14.7 (-16.3) | 15.4 (-8.6) | 45.9 (-4.8) | 29.7 (-0.9) | 51.0 (-4.0) | 52.4 (-9.3) | 36.8 (-7.8) |
| WizardMath | 13B | ✗ | 6.4 | 6.6 | 11.5 | 9.6 | 11.0 | 28.5 | 21.1 | 15.0 |
| ToRA-Code 🐯 | 13B | ✓ | **35.7 (+29.3)** | **31.1 (+24.5)** | **25.7 (+14.2)** | **55.6 (+46.0)** | **39.5 (+28.5)** | **58.7 (+30.2)** | **66.7 (+45.6)** | **48.1 (+33.1)** |
| w/o Shaping | 13B | ✓ | 32.8 (-2.9) | 26.0 (-5.1) | 24.0 (-1.7) | 52.6 (-3.0) | 38.4 (-1.1) | 55.6 (-3.1) | 61.2 (-5.5) | 44.6 (-3.5) |
| w/o Rationale | 13B | ✓ | 27.1 (-8.6) | 15.8 (-15.3) | 16.3 (-9.4) | 50.4 (-5.2) | 36.9 (-2.6) | 55.3 (-3.4) | 56.5 (-10.2) | 40.2 (-7.9) |
| ToRA-Code 🐯 | 34B | ✓ | **38.9** | **34.6** | **27.3** | **57.8** | **41.4** | **63.7** | **67.7** | **50.8** |
| w/o Shaping | 34B | ✓ | 34.0 (-4.9) | 29.9 (-4.7) | 24.6 (-2.7) | 55.6 (-2.2) | 41.6 (+0.2) | 63.8 (+0.1) | 61.4 (-6.3) | 47.4 (-3.4) |
| w/o Rationale | 34B | ✓ | 28.3 (-10.6) | 15.8 (-18.8) | 18.0 (-9.3) | 52.4 (-5.4) | 40.7 (-0.7) | 58.6 (-5.1) | 57.5 (-10.2) | 41.9 (-8.9) |
| WizardMath | 70B | ✗ | 9.1 | 13.4 | 16.9 | 16.5 | 19.2 | 42.7 | 35.0 | 24.1 |
| ToRA 🐯 | 70B | ✓ | **37.1 (+28)** | **30.4 (+17)** | **30.1 (+13.2)** | **54.6 (+38.1)** | **40.3 (+21.1)** | **64.9 (+22.2)** | **66.6 (+31.6)** | **49.7 (+25.6)** |
| w/o Shaping | 70B | ✓ | 33.8(-3.3) | 28.9(-1.5) | 27.1(-3) | 53.0(-1.6) | 38.0(-2.3) | 62.2(-2.7) | 64.2(-2.4) | 47.3(-2.4) |
| w/o Rationale | 70B | ✓ | 26.7(-10.4) | 14.7(-15.7) | 20.3(-9.8) | 48.9(-5.7) | 39.2(-1.1) | 59.8(-5.1) | 57.6(-9) | 41.5(-8.2) |

## 3.4 MAIN RESULTS

Table 2 presents the results of ToRA on 10 mathematical datasets, highlighting the following salient observations: **(1)** Using interleaved formatting and output space shaping, ToRA consistently surpasses prior state-of-the-art open-source models across all scales, achieving 13% to 19% absolute improvements across 10 tasks. **(2)** ToRA-70B substantially outperforms ChatGPT with both CoT and PAL prompting on GSM8k (84.3% vs. 80.4%) and MATH (49.7% vs. 38.7%), while ToRA-CODE-34B is competitive with GPT-4 solving competition-level MATH dataset with code (50.8% vs. 51.8%). **(3)** The accuracy of ToRA-CODE is about 5% higher than ToRA of the same size, demonstrating that continued training on code data significantly benefits program-based tool use. **(4)** While rationale-based fine-tuning negatively affects out-of-distribution generalization, ToRA displays superior generalization. For instance, WizardMath-70B underperforms the base model on TabMWP (49.8% vs. 57.5%), while ToRA-70B effectively generalizes to this tabular reasoning task (74.0%). **(5)** ToRA attains fast zero-shot inference speed, averaging 1.02 tool interaction rounds per problem, while effectively addressing problems that require interactive tool utilization.

## 3.5 ABLATION STUDY

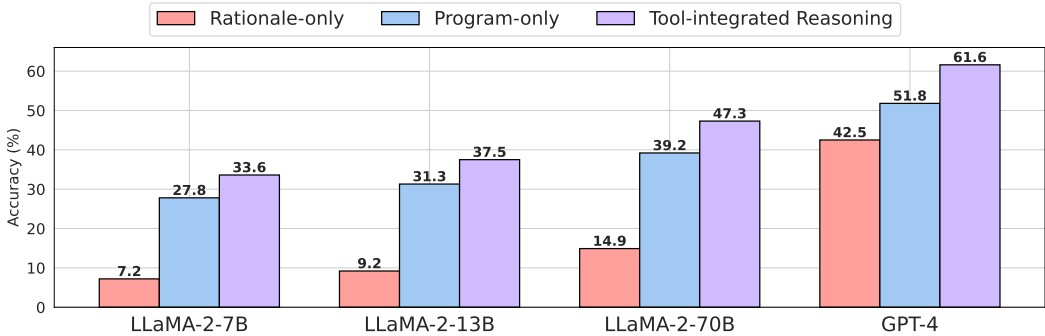

Figure 4: Comparison of three formats: (1) Rationale-only: step-by-step natural language reasoning like CoT; (2) Program-only: solving problems with programs like PAL; (3) Tool-integrated Reasoning used by ToRA: interweaving rationale and program execution to solve problems. We evaluated GPT-4 with few-shot prompting. We trained LLaMA-2 models to reason in the three types of formats, respectively. For a fair comparison, we *do not apply output space shaping* for all LLaMA-2 models.

### 3.5.1 COMPARISONS OF FORMATTING

To evaluate the efficacy of the reasoning format adopted by TORA which interleaves rationales with programs, we compared it with Rationale-only and Program-only formats using GPT-4 and LLaMA-2 trained with the same size of data from MATH. As shown in Fig 4, the TORA method consistently surpasses Rationale-only and Program-only approaches. Remarkably, using LLaMA-2, the TORA method achieves substantial improvements of 29.0% and 6.7% over Rationale-only and Program-only, respectively. With the closed-source GPT-4, the improvements are 19.1% and 9.8%, respectively. This emphasizes the effectiveness of integrating natural language rationales with programs.

### 3.5.2 EFFECTS OF OUTPUT SPACE SHAPING

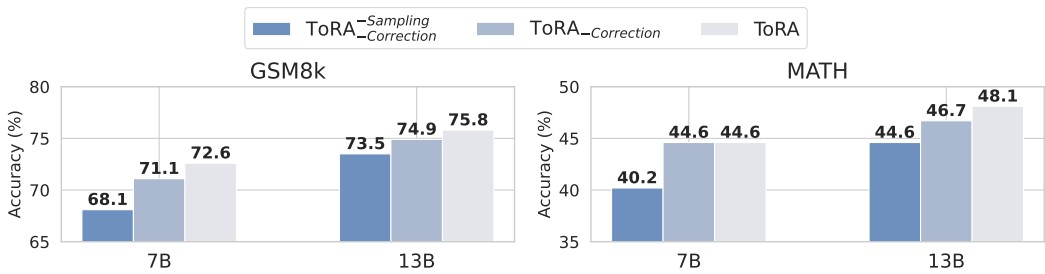

Figure 5: Ablation on output space shaping strategies using CodeLLaMA: (1) $\text{TORA}_{-\text{Correction}}^{-\text{Sampling}}$ is trained on TORA-CORPUS without shaping. (2) $\text{TORA}_{-\text{Correction}}$ employs only the sampling strategy for shaping, trained with up to 4 additional valid trajectory samples per problem. (3) TORA utilizes both the sampling and correction, also trained with up to 4 additional trajectories per problem.

We assess the effectiveness of the output space shaping strategies presented in Section 2.3, specifically sampling and correction. As shown in Fig 5 and Table 3: (1) Output space shaping yields a considerable average improvement of 3.4% and 4.0% absolute for GSM8k and MATH, respectively, with greater benefits for smaller models; (2) Applying the sampling strategy results in a 2.7% absolute improvement on average, while additionally incorporating correction offers a more substantial boost of up to 4.5%, without using more training data; (3) Output space shaping benefits even the largest model TORA-70B, with a notable improvement from 47.3% to 49.7% on MATH. These findings highlight the effectiveness of our shaping strategies across different model sizes and datasets.

### 3.6 ANALYSIS

We investigate the benefits, detailed patterns, and remaining challenges of tool interaction for mathematical reasoning on the challenging MATH dataset. Performance breakdowns on all subtopics of MATH are reported in Table 3.

**Benefits from Tool-Integration for MATH Sub-topics** As shown in Table 3, TORA outperforms WizardMath by around 45% in Algebra and Number Theory, which is attributed to stimulating and shaping tool-use behavior. Problems from the two sub-topics typically need intricate computation and data manipulation. Algebra mainly focuses on solving equations and application problems, while many Number Theory problems can be tackled using brute-force approaches through code.

**Patterns of Library Usage for Problem Solving** Fig 6 presents the most frequently used libraries for different sub-topics and the corresponding accuracies of their solutions. Tool-use behavior on different mathematical areas demonstrates distinct patterns. `sympy` and its internal `solvers` are primarily employed for algebra-related topics. Precalculus exhibits extensive matrix operations via `matrices`, resulting in a high accuracy. Number Theory depends on `algorithms` like `gcd` and `lcm`. Geometry mainly uses the `rational` library for fraction-based computations, while the application of other tools is limited, signifying the potential for improvement.

**Detailed Impact of Rationale on Different Topics** Table 3 shows that using an interleaved format, in contrast to merely writing the program, leads to significant improvements across all subtopics, especially in Precalculus, Algebra, and Geometry, where notable increases range from 8.6% to 18.8%.

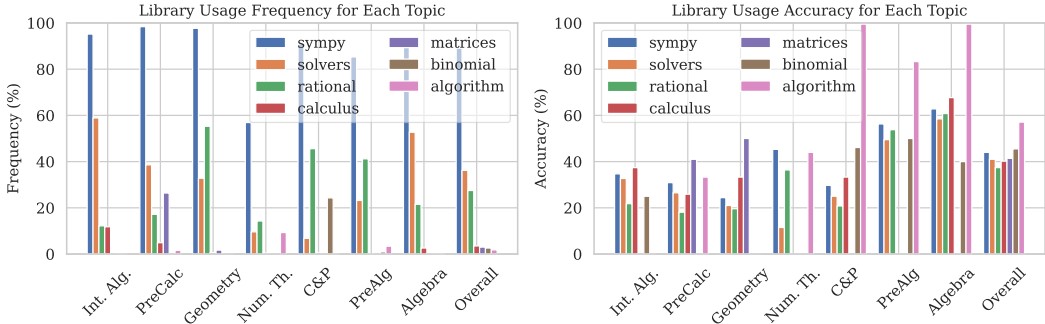

Figure 6: Library usage frequency and accuracy on each sub-topic of MATH.

Appendix F.1 provides representative examples demonstrating how the rationale aids in planning, multi-round self-correction, and finalizing answers.

Table 4: The failure modes of the TORA on MATH, and their corresponding percentages in random samples analyzed by humans. We include specific examples of each failure mode in Appendix F.

| Error Type | Definition | % | Examples |
|---|---|---|---|
| Reasoning Error | Mistakes due to incorrect reasoning steps or missing conditions. | 38% | Ex. 5 |
| Hallucination | Fabrication of numbers or answers. | 5% | Ex. 6 |
| Diagram Understanding | Misinterpretation of the input diagram. | 21% | Ex. 7 |
| Inappropriate Tool Use | Incorrect use of external tools, especially when the problem can't be solved directly with libraries. | 10% | Ex. 8 |
| Syntax Error | Persistent syntax errors despite multiple correction attempts. | 9% | Ex. 9 |
| Runtime Error | Errors during program execution, unresolved by retrying. | 9% | Ex. 10 |
| Rationale-only Error | Cannot be formalized into a program and the rationale is incorrect. | 3% | Ex. 11 |
| False Negative | Correct answers that don't fully match the ground truth. | 5% | Ex. 12 |

**Remaining Challenges in Mathematical Reasoning for TORA** To better understand the failure modes and remaining challenges, we manually annotated 100 randomly selected trajectories from the MATH test set, identifying and categorizing their failure modes. The results are shown in Table 4: Primarily, incorrect reasoning steps constitute the primary source of errors for ToRA on complex math reasoning tasks (38%), with some hallucination issues also evident during problem interpretation and answer finalization (5%). Secondly, the misinterpretation of input diagrams contributes significantly to the error rate (21%). This is particularly noticeable in Geometry, Precalculus, and Intermediate Algebra. The diagrams in the MATH dataset are usually detailed in text using the Asymptote language (Hendrycks et al., 2021), thus making it challenging for TORA to comprehend diagrams purely from textual descriptions. Thirdly, issues with tool usage include Inappropriate Tool Usage (10%), Syntax Error (9%), and Runtime Error (9%). These problems frequently arise when TORA fails to use tools correctly after several corrections or attempts. There are certain inputs that fail to formalize well as programs (3%), which require abstract reasoning rather than computation. Finally, we also found that there are false negatives when using automatic indicators, i.e., correct predictions that are misjudged as wrong, but the proportion is relatively small (5%).

## 4 CONCLUSION

This paper presents TORA, a series of novel Tool-integrated Reasoning Agents that synergistically combines natural language rationale with program-based tool-use for mathematical problem solving. Our approach demonstrates the potential of integrating external tools in the reasoning process, enabling language models to effectively tackle complex quantitative tasks. TORA achieves state-of-the-art performance on 10 diverse mathematical reasoning tasks, substantially outperforming existing rationale-based and program-based approaches. Furthermore, our systematic analysis of the benefits and remaining challenges of tool interaction provides valuable insights for future research, contributing to the development of more advanced and versatile reasoning agents.

AUTHOR CONTRIBUTIONS

Zhibin Gou proposed the interleaved tool-use format of TORA and curated TORA-CORPUS dataset, implemented the training and evaluation pipeline, conducted experiments and analysis on all datasets, implemented baselines, and was a main contributor to the paper writing. Zhihong Shao proposed the project, conducted preliminary experiments, proposed and implemented the training and evaluation pipelines, proposed and trained all TORA models with output space shaping as well as TORA variants in the ablation study, designed and oversaw experimental analysis, and contributed to many parts of the paper writing. Yeyun Gong, Yelong Shen, Yujiu Yang, Minlie Huang, Nan Duan, and Weizhu Chen provided research mentorship, oversaw project coordination, and advised and contributed to many parts of the writing.

ACKNOWLEDGMENTS

Zhibin Gou and Yujiu Yang were supported by the National Natural Science Foundation of China (Grant No. U1903213) and the Shenzhen Science and Technology Program (JSGG20220831110203007). Zhihong Shao and Minlie Huang were supported by the NSFC projects (Key project with No. 61936010 ), and were also supported by the National Science Foundation for Distinguished Young Scholars (with No. 62125604).

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

CONTENTS

## A   RELATED WORKS

**Mathematical Reasoning** Recent research has greatly improved reasoning in LLMs with step-by-step natural language reasoning (Polu & Sutskever, 2020; Wei et al., 2022; Zhou et al., 2023b; Zhu et al., 2023; Huang et al., 2022; Liang et al., 2023). However, natural language reasoning struggles with complex computations and symbolic manipulations. To overcome the limitations, recent research has exploited tools like calculators (Cobbe et al., 2021; Shao et al., 2022), code interpreters (Mishra et al., 2022), and symbolic solvers (Zhang et al., 2023). Program-based methods (Gao et al., 2022; Chen et al., 2022; Shao et al., 2023a) transform reasoning tasks into program synthesis tasks, thus offering complementary advantages over natural language reasoning, but they face challenges in nuanced reasoning, planning, and error handling (Gou et al., 2023), where natural language reasoning should be more suitable.

**Tool-Augmented Language Models** Augmenting LLMs with tools can largely alleviate LLMs' limitations and improve reasoning and generation performance (Parisi et al., 2022; Mialon et al., 2023; Yao et al., 2023). Recent work demonstrates the benefits of integrating retrievers (Borgeaud et al., 2022; Shao et al., 2023b), search engines (Nakano et al., 2021), and multi-tool approaches (Schick et al., 2023; Paranjape et al., 2023; Gou et al., 2023) to improve generation.

**Knowledge Distillation** Knowledge distillation (KD) transfers knowledge from teacher models to student models (Buciluǎ et al., 2006; Hinton et al., 2015). Using LLM-generated trajectories for fine-tuning is a form of KD (Fu et al., 2023; Taori et al., 2023; Peng et al., 2023; Ho et al., 2023). Our proposed TORA shows that learning interactive tool-use trajectories is a promising direction to adapt language models to reasoning tasks.

## B   EVALUATION DATASETS

We present statistics and examples of the ten evaluation datasets in Table 5.

Table 5: Statistics and examples of the 10 evaluation datasets. In the main result table, we present the average accuracy of SingleEq, SingleOp, AddSub, and MultiArith under the collective name MAWPS.

| Dataset | OOD? | #Samples | Example Problem |
|---|---|---|---|
| GSM8k (Cobbe et al., 2021) | IND | 1319 | The ice cream parlor was offering a deal, buy 2 scoops of ice cream, get 1 scoop free. Each scoop cost $1.50. If Erin had $6.00, how many scoops of ice cream should she buy? |
| MATH (Hendrycks et al., 2021) | IND | 5000 | For a constant $c$, in cylindrical coordinates $(r, \theta, z)$, find the shape described by the equation $$z = c.$$ (A) Line (B) Circle (C) Plane (D) Sphere (E) Cylinder (F) Cone. Enter the letter of the correct option. |
| GSM-Hard (Gao et al., 2022) | OOD | 1319 | Jean has 30 lollipops. Jean eats 8714250 of the lollipops. With the remaining lollipops, Jean wants to package 8714250 lollipops in one bag. How many bags can Jean fill? |
| SVAMP (Patel et al., 2021) | OOD | 1000 | During summer break 819058 kids from Lawrence county go to camp and the other 668278 kids stay home. How many more kids spent their summer break at the camp compared to those who stayed home? |
| ASDiv (Miao et al., 2020) | OOD | 2215 | Mrs. Hilt saw an iPod for sale. The price tag said the iPod cost $128, but a sign announced that it was on sale for "35% off." How much would the iPod cost after the discount? |
| TabMWP (Lu et al., 2023) | OOD | 1000 | Stem / Leaf table: 2 \| 3, 6, 7, 8, 8; 3 \| 0, 7, 9; 4 \| 1, 5; 5 \| ; 6 \| 2, 3, 3, 4, 8, 8; 7 \| 3, 4, 4, 7, 9; 8 \| 5, 5. Read the table regarding "eight lifting results (lbs)". Mr. Morrison, a P.E. teacher, wrote down how much weight each of his students could lift. How many people lifted at least 28 pounds? |
| SingleEq (Koncel-Kedziorski et al., 2016) | OOD | 508 | Alyssa's dog had puppies. She gave 7 to her friends.She now has 5 puppies left. How many puppies did she have to start with? |
| SingleOp (Koncel-Kedziorski et al., 2016) | OOD | 562 | Rachel removes 47 bottle caps from a jar. There were originally 87 bottle caps in the jar. How many bottle caps are left in the jar? |
| AddSub (Koncel-Kedziorski et al., 2016) | OOD | 395 | Sam went to 14 football games this year. He went to 29 games last year. How many football games did Sam go to in all? |
| MultArith (Koncel-Kedziorski et al., 2016) | OOD | 600 | Paige had 43 math problems and 12 science problems for homework. If she finished 44 of the problems at school, how many problems did she have to do for homework? |

Table 6: Accuracies of ChatGPT and GPT-4 on the MATH dataset, with breakdown w.r.t. different mathematical subjects. We apply PAL prompting and the Tool-integrated Reasoning method used by ToRA to the two closed-source models.

| Model | Tool | Intermediate Algebra | Precalculus | Geometry | Number Theory | Counting & Probability | Prealgebra | Algebra | Overall |
|---|---|---|---|---|---|---|---|---|---|
| | | | | | Test Set | | | | |
| ChatGPT (PAL) 🐍 | ✓ | 18.5 | 19.2 | 23.2 | 48.5 | 43.0 | 62.7 | 45.4 | 38.7 |
| GPT-4 (PAL) 🐍 | ✓ | 32.8 | 29.3 | 38.0 | 58.7 | 61.0 | 73.9 | 59.1 | 51.8 |
| GPT-4 (Tool-integrated Reasoning) | ✓ | 40.0 | 37.2 | 44.1 | 68.9 | 67.3 | 82.2 | 75.8 | 61.6 |
| | | | | | Training Set | | | | |
| GPT-4 (Tool-integrated Reasoning) | ✓ | 51.0 | 51.5 | 42.5 | 77.4 | 72.2 | 89.8 | 85.1 | 64.3 |
| w/ best@10 | ✓ | 72.9 | 70.0 | 58.9 | 91.6 | 81.7 | 95.5 | 96.3 | 83.1 |

## C  ADDITIONAL EXPERIMENTS AND ANALYSIS

### C.1  ACCURACIES OF CLOSED-SOURCE MODELS ON MATH

Table 6 presents the detailed accuracies of GPT-4 on the MATH dataset. The Tool-integrated Reasoning method used by ToRA significantly outperforms PAL prompting when directly applied to the closed-source GPT-4, further demonstrating the benefits of synergizing natural language reasoning and program-based tool use.

### C.2  EFFECTS OF # VALID TRAJECTORIES FOR OUTPUT SPACE SHAPING

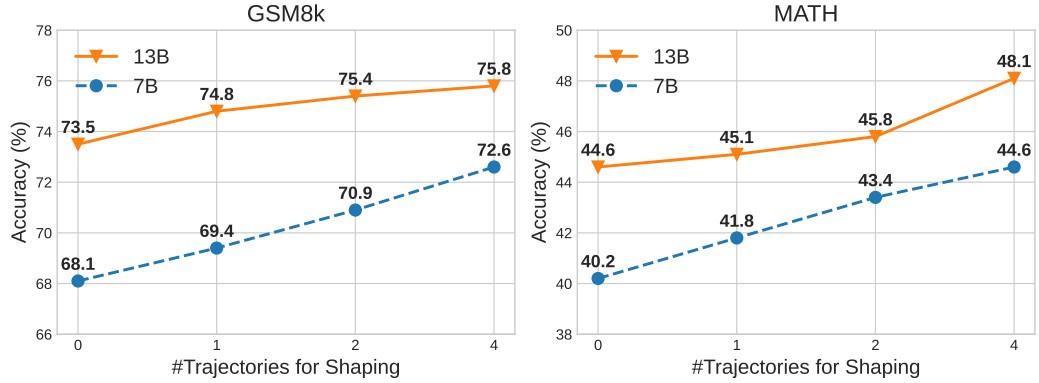

Figure 7: Effects of using different numbers of additional valid trajectories per question for output space shaping.

As shown in Fig 7, it is beneficial to increase the number of additional valid trajectories for output space shaping.

### C.3  IMPACT OF OUTPUT SPACE SHAPING IN RELATION TO QUESTION DIFFICULTY

We compare the effects of output space shaping on MATH problems of different difficulty levels (from level 1 to level 5) in Figure 8, and present the statistics of MATH problems at different levels in Table 7. As can be seen:

- Across these different difficulty levels and model sizes, output space shaping generally brings a significant improvement of 4.0% on average across different model sizes.

- Output space shaping brings significant improvements for difficult, long problems. E.g., with ToRA-Code-13B, shaping does not significantly improve level 1 to level 2 problems, but it brings a substantial improvement of 5.4% to 5.7% for level 3 to level 5 problems.

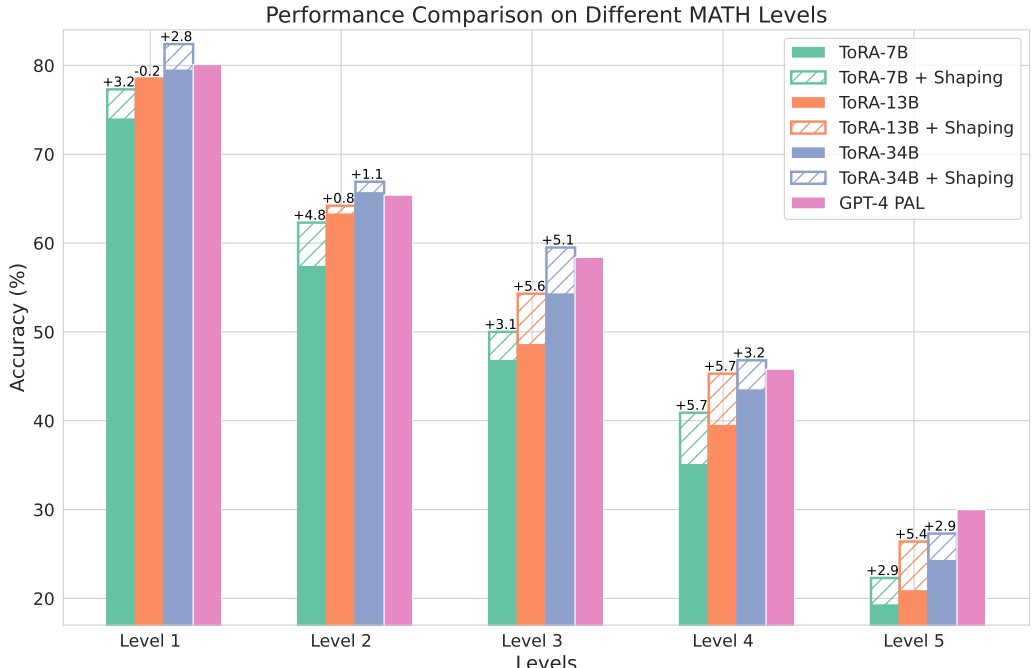

Figure 8: Impact of Output Space Shaping in Relation to Question Difficulty.

- After using shaping, TORA-CODE-34B outperforms GPT-4 PAL on problems from Level 1 to Level 4, but there is still a gap at Level 5 (27.3% vs. 30.0%). These problems are usually longer (average about 248.4 characters), require more reasoning steps (>1,000 characters) to solve, and more often include diagram inputs (about 20%). These observations may guide future work to focus more on solving these more difficult problems.

Table 7: Statistics of MATH problems at different levels. Average Answer Length indicates the average length of TORA outputs; Training query coverage indicates the proportion of queries with at least one valid trajectory in TORA-CORPUS relative to the total queries in the original dataset.

|                         | Level 1 | Level 2 | Level 3 | Level 4 | Level 5 |
|-------------------------|---------|---------|---------|---------|---------|
| # Test Samples          | 437     | 894     | 1131    | 1214    | 1324    |
| Avg Question Length     | 123.8   | 150.9   | 169.1   | 203.0   | 248.4   |
| Avg Answer Length       | 503.1   | 655.8   | 751.2   | 881.6   | 1083.8  |
| Training query coverage | 97.7%   | 91.6%   | 86.5%   | 81.3%   | 68.0%   |

# D   DETAILED INFORMATION OF TORA-CORPUS

We provide a more detailed introduction to the data construction process, quality control, and data statistical information, beyond Sec. 2.2.

**Data Format and Quality Control**   In our preliminary experiments, we found that the tool-integrated reasoning trajectory format generated by zero-shot prompting was somewhat chaotic. Therefore, we designed a few-shot prompting to control the reasoning format, which effectively improved data quality. On the other hand, we increased the annotation success rate by sampling, ensuring more comprehensive coverage of the training query.

**Data Filtering Process**   For the data constructed, we filtered out paths that produced incorrect answers by matching them with standard answers. To prevent the model from learning incorrect

Table 8: Accuracy of TORA-CORPUS on GSM8k and MATH training set. TORA-CORPUS-Greedy uses only the greedy trajectories, while ToRA-Corpus-16k combines sampled trajectories.

| | GSM8k | MATH | | | | | | | |
|---|---|---|---|---|---|---|---|---|---|
| | **All** | **All** | **Intermediate Algebra** | **Precalculus** | **Geometry** | **Number Theory** | **Counting & Probability** | **Prealgebra** | **Algebra** |
| TORA-CORPUS-Greedy | 94.4 | 64.3 | 51.0 | 51.5 | 70.0 | 77.4 | 72.2 | 89.8 | 85.1 |
| TORA-CORPUS-16k | 98.2 | 83.1 | 72.9 | 70.0 | 58.9 | 91.6 | 81.7 | 95.5 | 96.3 |

Table 9: Statistics of TORA-CORPUS-16k

| | **GSM8k** | **MATH** | **Total** |
|---|---|---|---|
| # Train Samples | 7,657 | 7,881 | 15,538 |
| Avg Question Length | 236 | 189 | 211 |
| Avg Trajectory Length | 678 | 704 | 691 |
| Min Trajectory Length | 218 | 119 | 119 |
| Max Trajectory Length | 1,713 | 2,486 | 2,486 |

intermediate reasoning processes, we further filtered out data samples with intermediate program execution errors.

**Dataset Statistics** In Table 8, we compared the annotation accuracy (i.e., sample coverage) of the training set on GSM8k, MATH, and MATH subtopics of TORA-CORPUS-Greedy using only the greedy trajectories, and TORA-CORPUS-16k combined with sampled trajectories. Furthermore, in Table 9, we reported the statistical data of TORA-CORPUS-16k, such as the number of samples, average question length, average, minimum, and maximum trajectory length, as shown in the following tables.

**Rationale as Hints** As described in Section 2.2, we annotated interactive tool-use trajectories for the training questions from MATH with GPT-4. GPT-4 achieves a success rate below 65% using greedy decoding. As MATH was originally annotated with natural language rationales, to improve the annotation success rate, we tried to provide GPT-4 with the human rationales as hints (Zelikman et al., 2022). However, when using this method, GPT-4 tends to replicate the hints and ignore tool-use outputs especially when the outputs are inconsistent with the hints, thus failing to produce high-quality trajectories. Hence, we deferred the utilization of the already-annotated natural language rationales for future investigations. Instead, we employed nucleus sampling to recall valid trajectories for questions that remained unsolved through greedy decoding. This approach significantly boosted annotation accuracy to 83.1%.

# E PROMPTS

We present instructions and example few-shot prompts of Tool-integrated Reasoning for querying GPT-4.

Listing 1: Prompts of Tool-integrated Reasoning for GPT-4.

```
Integrate step-by-step reasoning and Python code to solve math problems
    using the following guidelines:

- Analyze the question and write functions to solve the problem; the
    function should not take any arguments.
- Present the final result in LaTeX using a `\boxed{}` without any units.
- Utilize the `pi` symbol and `Rational`` from Sympy for $\pi$ and
    fractions, and simplify all fractions and square roots without
    converting them to decimal values.

Here are some examples you may refer to:

---

Question: Convert the point $(0, -3 \sqrt{3}, 3)$ in rectangular
    coordinates to spherical coordinates. Enter your answer in the form $
    (\rho,\theta,\phi),$ where $\rho > 0,$ $0 \le \theta < 2 \pi,$ and $0
     \le \phi \le \pi.$

Solution:
To convert the point $(x, y, z)$ in rectangular coordinates to spherical
    coordinates $(\rho, \theta, \phi)$, we use the following formulas:
1. $\rho = \sqrt{x^2 + y^2 + z^2}$
2. $\theta = \arctan{\frac{y}{x}}$
3. $\phi = \arccos{\frac{z}{\rho}}$

Let's write a function:

```python
from sympy import sqrt, atan2, acos, pi

def rectangular_to_spherical():
    x, y, z = 0, -3*sqrt(3), 3
    rho = sqrt(x**2 + y**2 + z**2)
    theta = atan2(y, x)
    phi = acos(z/rho)
    return rho, theta, phi

spherical_coordinates = rectangular_to_spherical()
print(spherical_coordinates)
```
```output
(6, -pi/2, pi/3)
```

The required form of the answer is $(\rho, \theta, \phi)$, where $\rho >
    0$, $0 \le \theta < 2\pi$, and $0 \le \phi \le \pi$. Since our
    initial output has $\theta = -\pi/2$, which doesn't meet the
    condition for $\theta$, we add $2\pi$ to adjust its value.

Hence, the spherical coordinates for the point $(0, -3\sqrt{3}, 3)$ are $
    \boxed{(6, \frac{3\pi}{2}, \frac{\pi}{3})}$.

---

Question: Compute $1011_2 + 101_2 - 1100_2 + 1101_2$. Express your answer
     in base 2.
```

Solution:
Let's write a single python function for binary computation:

```python
def binary_sum_diff():
    # Convert binary to decimal
    num1 = int("1011", 2)
    num2 = int("101", 2)
    num3 = int("1100", 2)
    num4 = int("1101", 2)

    # Perform the operations
    result = num1 + num2 - num3 + num4

    # Convert the result back to binary
    result_binary = format(result, "b")

    return result_binary

result = binary_sum_diff()
print(result)
```
```output
10001
```
The answer expressed in base 2 is $\boxed{10001_2}$.

---

Question: Find the matrix that corresponds to projecting onto the vector
$\begin{pmatrix} 2 \\ -3 \end{pmatrix}.$

Solution:
```python
from sympy import Matrix, simplify

def projection_matrix():
    v = Matrix([2, -3])
    P = (v * v.T) / (v.T * v)[0]

    # Simplify the elements
    P = P.applyfunc(simplify)

    return P

result = projection_matrix()
print(result)
```
```output
Matrix([[4/13, -6/13], [-6/13, 9/13]])
```

The matrix is $\boxed{\begin{pmatrix} \frac{4}{13} & -\frac{6}{13} \\ -\frac{6}{13} & \frac{9}{13} \end{pmatrix}}.$

---

Question: Solve the inequality
\[\frac{x^2 - 25}{x + 5} < 0.\]

Solution:
Let's use `sympy` to solve the inequality:

```python
from sympy import symbols, simplify
```

```
from sympy.solvers.inequalities import solve_univariate_inequality
from sympy.core.relational import StrictLessThan

def solve_inequality():
    x = symbols('x')
    expression = (x**2 - 25) / (x + 5)
    inequality = StrictLessThan(expression, 0)
    solution = solve_univariate_inequality(inequality, x, relational=False
        )
    simplified_solution = simplify(solution)

    return simplified_solution

result = solve_inequality()
print(result)
```
```output
Union(Interval.open(-oo, -5), Interval.open(-5, 5))
```
The answer is $\boxed{(-\infty,-5)\cup(-5,5)}$

---

## F  EXAMPLES

### F.1  SUCCESS CASES

Listing 2: Success case for TORA: **Rationale for planning**.

Question: What real values of $x$ are not in the domain of $f(x)=\frac
    {1}{|x^2-6x+8|+|x^2+x-6|}$?

Solution:
The domain of the function is all real numbers, so we need to find the
    values of x for which the denominator of the function is zero.
The denominator of the function is the sum of the absolute values of the
    two terms inside the absolute value.
Each of these terms is a quadratic in x, and the roots of a quadratic
    equation are real unless the equation is identically zero.
So, we need to find the values of x for which the two quadratic equations
     are identically zero.

```python
from sympy import symbols, Abs, solveset, S
from sympy.core.relational import Ne

def non_domain_values():
    x = symbols('x')
    # These are the values of x for which the function is not defined
    roots1 = solveset(Ne(x**2 - 6*x + 8, 0), x, domain=S.Reals)
    roots2 = solveset(Ne(x**2 + x - 6, 0), x, domain=S.Reals)
    # The roots of a quadratic equation are always in the domain of the
        function, so we need to find the values of x that are not in the
        union of the roots
    non_domain_values = S.Reals - (roots1.union(roots2))

    return non_domain_values

result = non_domain_values()
print(result)
```
```output
{2}
```

```
```
The real value of $x$ that is not in the domain of the function is $\
    boxed{2}$.
```

Listing 3: Success case for TORA: **Self-Correcting Errors with tool feedback**.

```
Question: The difference between two perfect squares is 133. What is the
    smallest possible sum of the two perfect squares?

Solution:
We can solve this problem by noting that any perfect square can be
    written as $(a + b)^2$ or $(a - b)^2$ for some integers $a$ and $b$.
     So we can rewrite the question as the following problem: Find the
    minimum possible value for $(a + b)^2 + (a - b)^2$ given that $(a + b)
    ^2 - (a - b)^2 = 133$.

```python
from sympy import symbols, Eq, solve

def min_sum_perfect_squares():
    """
    Find the smallest possible sum of two perfect squares
    given that the difference between them is 133.

    Returns:
        Integer: The minimum possible sum of the two perfect squares.
    """
    a, b = symbols('a b')

    # Equation for the difference of the squares
    diff_eq = Eq((a + b)**2 - (a - b)**2, 133)

    # Solve the equation for b
    b_solutions = solve(diff_eq, b)

    # Calculate the sum of the squares for each solution of b
    sum_squares = [(a + b_sol)**2 + (a - b_sol)**2 for b_sol in
        b_solutions]

    # Find the minimum possible sum of the squares
    min_sum_squares = min(sum_squares)

    return min_sum_squares

result = min_sum_perfect_squares()
print(result)
```
```output
(a - 133/(4*a))**2 + (a + 133/(4*a))**2
```

The problem seems to be more complex than anticipated and the solution
    was incorrect.
Since the difference between two squares can be factored, we can write:

$x^2 - y^2 = (x + y)(x - y)$

$133$ has two pairs of factors: 1 and 133, and 7 and 19.

So, either $x+y = 133$ and $x-y = 1$, or $x+y = 19$ and $x-y = 7$.

- For (133, 1), $x = \frac{133 + 1}{2} = 67$ and $y = \frac{133 - 1}{2} =
    66$.
- For (19, 7), $x = \frac{19 + 7}{2} = 13$ and $y = \frac{19 - 7}{2} = 6$.
```

```
(13, 6) result in the smallest sum of the two perfect squares.

So, let's modify the previous solution:

```python
def min_sum_perfect_squares():

    min_squares = [13**2, 6**2]
    min_sum_squares = sum(min_squares)

    return min_sum_squares
```
```output
205
```

The smallest possible sum of the two perfect squares is $\boxed{205}$.
```

Listing 4: Success case for TORA: **Finalizing answer**.

```
Question:
Compute \[\sum_{k = 1}^\infty \frac{k + 2}{k! + (k + 1)! + (k + 2)!}.\]

Solution:
```python
from sympy import symbols, Sum, factorial, oo, simplify

def infinite_sum():
    expr = (k + 2) / (factorial(k) + factorial(k + 1) + factorial(k + 2))
    sum_expr = Sum(expr, (k, 1, oo))
    result = sum_expr.doit()

    return result

result = infinite_sum()
print(result)
```
```output
Sum((k + 2)/(factorial(k) + factorial(k + 1) + factorial(k + 2)), (k, 1,
    oo))
```
The sum does not converge to a simple expression, so the answer is $\
    boxed{\sum_{k = 1}^\infty \frac{k + 2}{k! + (k + 1)! + (k + 2)!}}$.
```

## F.2 FAILURE CASES

Listing 5: Failure case for TORA: **Reasoning Error**.

```
Idx: 4289
Type: Prealgebra
Level: Level 2
Question: 'What is the smallest positive multiple of $13$ that is greater
    than $500?$'
Solution:
 ```python
def smallest_positive_multiple():
    """What is the smallest positive multiple of $13$ that is greater than
        $500?$"""
    # Start from 500 and keep adding 13 until we get a number that is
        greater than 500
    num = 500
    while num <= 500:
```

```
      num += 13

   return num

result = smallest_positive_multiple()
print(result)
```
```output
513
```
The smallest positive multiple of $13$ that is greater than $500$ is $\
    boxed{513}$.

Execution: None
Prediction: '513'
Ground Truth: '507'
Score: [False]
Reference CoT: 'Dividing $500$ by $13$ gives a quotient $38$ with a
    remainder of $6$. In other words, \\[500 \\div 13=13 \\cdot 38+6.\\]
     So, the smallest positive multiple of $13$ that is greater than $500$
     is \\[13\\cdot39=\\boxed{507}.\\]'
```

Listing 6: Failure case for TORA: **Hallucination**.

```
Idx: 2775
Type: Intermediate Algebra
Level: Level 3
Question: 'Find the eighth term of the sequence $1440, 1716, 1848, \\
    ldots$, whose terms are formed by multiplying the corresponding terms
     of two arithmetic sequences.'
Solution:
 ```python
from sympy import symbols, Rational, simplify

def eighth_term():
   """Find the eighth term of the sequence $1440, 1716, 1848, \ldots$,
       whose terms are formed by multiplying the corresponding terms of
       two arithmetic sequences."""
   a1, d1 = 1440, 36
   a2, d2 = 16, 2

   # Calculate the eighth term of each sequence
   term1 = a1 + 7*d1
   term2 = a2 + 7*d2

   # Multiply the terms
   result = term1 * term2

   return result

result = eighth_term()
print(result)
```
```output
50760
```
The eighth term of the sequence is $\boxed{50760}$.

Execution: None
Prediction: '50760'
Ground Truth: '348'
Score: [False]
Reference CoT: 'If $a_n =an + b$ and $b_n = cn + d$ are two arithmetic
    sequences, then their term-wise product takes the form \\[a_nb_n = (
    an+b)(cn+d) = An^2 + Bn + C,\\]where $A, B, C$ are constants.
```

```
Therefore, some sequence $x_n = An^2 + Bn + C$ has $x_0 = 1440,$ $x_1
    = 1716,$ and $x_2 = 1848.$ Then we have the equations \\[\\begin{
    aligned} C &= 1440, \\\\ A+B+C&=1716,\\\\ 4A+2B+C&=1848. \\end{
    aligned}\\]Subtracting $C=1440$ from the second and third equations,
    we have $A+B=276$ and $4A+2B=408,$ or $2A+B=204.$ Then \\[A = (2A+B) -
    (A+B) = 204 - 276 = -72,\\]and so $B = 276-A=348.$ We conclude that
    the given sequence has the general formula \\[x_n = -72n^2 + 348n +
    1440.\\]Then the eighth term is \\[\\begin{aligned} x_7 &= -72 \\cdot
    7^2 + 348 \\cdot 7 + 1440 = \\boxed{348}. \\end{aligned}\\]'
```

Listing 7: Failure case for TORA: **Diagram Understanding**. Navigating the complexities of input diagrams is key to solving many problems in Geometry, Precalculus, and Intermediate Algebra. This ideal process encompasses not just visual modalities, but also involves interactions with diagrams, such as adding auxiliary lines. Nonetheless, the utility of computational tools like `SymPy` is, unfortunately, only marginally beneficial.

```
Question: A square has a side length of 10 inches. Congruent isosceles
    right triangles are cut off each corner so that the resulting octagon
     has equal side lengths. How many inches are in the length of one
    side of the octagon? Express your answer as a decimal to the nearest
    hundredth.

[asy]
size(150);
draw((0,0)--(10,0)--(10,10)--(0,10)--(0,0),linewidth(0.7));
draw((5*sqrt(2)/(1 + sqrt(2)),0)--(0,5*sqrt(2)/(1 + sqrt(2))),linewidth
    (0.7));
draw((10 - (5*sqrt(2)/(1 + sqrt(2))),0)--(10,5*sqrt(2)/(1 + sqrt(2))),
    linewidth(0.7));
draw((0,(10-(5*sqrt(2)/(1+sqrt(2)))))--(5*sqrt(2)/(1 + sqrt(2)),10),
    linewidth(0.7));
draw((10 - (5*sqrt(2)/(1 + sqrt(2))),10)--(10,10 - 5*sqrt(2)/(1 + sqrt(2)
    )),linewidth(0.7));
[/asy]

Solution:
The side length of the octagon is the hypotenuse of the right triangle
    formed by the side of the square and the diagonal of the octagon
The hypotenuse is the sum of the side length of the square and the length
     of the diagonal of the octagon
The length of the diagonal of the octagon is 5*sqrt(2)/(1 + sqrt(2))

```python
from sympy import sqrt, Rational

def octagon_side_length():
    square_side_length = 10
    octagon_side_length = square_side_length + 5*sqrt(2)/(1 + sqrt(2))

    return round(octagon_side_length, 2)

result = octagon_side_length()
print(result)
```
```output
12.93
```
The length of one side of the octagon is $\boxed{12.93}$ inches.

**Ground Truth**:
Recall that the ratio of the hypotenuse to the leg of an isosceles right
    triangle is $\sqrt{2}$. Looking at the removed triangles, we see that
     the three segments that make up the side of square are $s/\sqrt{2}$,
     $s$, and $s/\sqrt{2}$, where $s$ is the side length of the octagon.
```

```
    Setting the sum of these three side lengths equal to 10 inches, we
    find \begin{align*}
\frac{s}{\sqrt{2}}+s+\frac{s}{\sqrt{2}}&=10 \implies \\
s\left(\frac{1}{\sqrt{2}}+1+\frac{1}{\sqrt{2}}\right)&=10 \implies \\
s(\sqrt{2}+1)&=10 \implies \\
s&=\frac{10}{\sqrt{2}+1}=10(\sqrt{2}-1),
\end{align*} where we have rationalized the denominator twice: \[
\frac{1}{\sqrt{2}}=\frac{1}{\sqrt{2}}\cdot \frac{\sqrt{2}}{\sqrt{2}}=\
    frac{\sqrt{2}}{2},
\] and \[
\frac{10}{\sqrt{2}+1}=\frac{10}{\sqrt{2}+1}\cdot\frac{\sqrt{2}-1}{\sqrt
    {2}-1}=10(\sqrt{2}-1).
\] To the nearest hundredth, $s=\boxed{4.14}$ inches.

[asy]
size(150);
defaultpen(linewidth(0.7)+fontsize(10));
real s = 10/(1+sqrt(2));
draw((0,0)--(10,0)--(10,10)--(0,10)--(0,0));
draw((5*sqrt(2)/(1 + sqrt(2)),0)--(0,5*sqrt(2)/(1 + sqrt(2))));
draw((10 - (5*sqrt(2)/(1 + sqrt(2))),0)--(10,5*sqrt(2)/(1 + sqrt(2))));
draw((0,(10-(5*sqrt(2)/(1+sqrt(2)))))--(5*sqrt(2)/(1 + sqrt(2)),10));
draw((10 - (5*sqrt(2)/(1 + sqrt(2))),10)--(10,10 - 5*sqrt(2)/(1 + sqrt(2)
    )));

label("$s$",(10-s/(2*sqrt(2)),10-s/(2*sqrt(2))),SW);
label("$\displaystyle{\frac{s}{\sqrt{2}}}$",(10,10-s/(2*sqrt(2))),E);
label("$\displaystyle{\frac{s}{\sqrt{2}}}$",(10,s/(2*sqrt(2))),E);
label("$s$",(10,5),E);

draw(rightanglemark((10,0),(10,10),(0,10)));[/asy]
```

Listing 8: Failure case for TORA: **Inappropriate Tool Use**.

```
Idx: 2489
Type: Intermediate Algebra
Level: Level 5
Question: 'Let $z_1,$ $z_2,$ $z_3$ be complex numbers such that $|z_1| =
    2,$ $|z_2| = 3,$ and $|z_3| = 4.$ Find the largest possible value of\
    n\\[|z_1 - z_2|^2 + |z_1 - z_3|^2 + |z_2 - z_3|^2.\\]'
Solution:
 ```python
from sympy import symbols, simplify, Max
from sympy.functions import Abs

def max_expression_value():
    """Let $z_1,$ $z_2,$ $z_3$ be complex numbers such that $|z_1| = 2,$ $
        |z_2| = 3,$ and $|z_3| = 4.$ Find the largest possible value of
\\[|z_1 - z_2|^2 + |z_1 - z_3|^2 + |z_2 - z_3|^2.\]"""
    # Define the variables
    z1, z2, z3 = symbols('z1 z2 z3')

    # Define the expression
    expr = Abs(z1 - z2)**2 + Abs(z1 - z3)**2 + Abs(z2 - z3)**2

    # Define the conditions
    conditions = [Abs(z1) - 2, Abs(z2) - 3, Abs(z3) - 4]

    # Find the maximum value of the expression under the conditions
    max_value = Max(expr, *conditions)

    # Simplify the maximum value
    simplified_max_value = simplify(max_value)

    return simplified_max_value
```

```
result = max_expression_value()
print(result)
```
```output
Max(Abs(z1) - 2, Abs(z2) - 3, Abs(z3) - 4, Abs(z1 - z2)**2 + Abs(z1 - z3)
   **2 + Abs(z2 - z3)**2)
```
The largest possible value of $|z\_1 - z\_2|^2 + |z\_1 - z\_3|^2 + |z\_2 - z\_3
   |^2$ is $\boxed{25}$.

Execution: None
Prediction: '25'
Ground Truth: '87'
Score: [False]
Reference CoT: 'We have that\n\\begin{align*}\nz_1 \\overline{z}_1 &= |
   z_1|^2, \\\\\nz_2 \\overline{z}_2 &= |z_2|^2, \\\\\nz_3 \\overline{z}
   _3 &= |z_3|^2.\n\\end{align*}Likewise,\n\\begin{align*}\n&|z_1 - z_2
   |^2 + |z_1 - z_3|^2 + |z_2 - z_3|^2 \\\\\n&= (z_1 - z_2)(\\overline{
   z_1 - z_2}) + (z_1 - z_3)(\\overline{z_1 - z_3}) + (z_2 - z_3)(\\
   overline{z_2 - z_3}) \\\\\n&= (z_1 - z_2)(\\overline{z}_1 - \\
   overline{z}_2) + (z_1 - z_3)(\\overline{z}_1 - \\overline{z}_3) + (
   z_2 - z_3)(\\overline{z}_2 - \\overline{z}_3) \\\\\n&= z_1 \\overline{
   z}_1 - z_1 \\overline{z}_2 - \\overline{z}_1 z_2 + z_2 \\overline{z}
   _2 + z_1 \\overline{z}_1 - z_1 \\overline{z}_3 - \\overline{z}_1 z_3 +
    z_1 \\overline{z}_3 + z_2 \\overline{z}_3 - z_2 \\overline{z}_3 - \\
   overline{z}_2 z_3 + z_2 \\overline{z}_3 \\\\\n&= 2|z_1|^2 + 2|z_2|^2 +
    2|z_3|^2 - (z_1 \\overline{z}_2 + \\overline{z}_1 z_2 + z_1 \\
   overline{z}_3 + \\overline{z}_1 z_3 + z_2 \\overline{z}_3 + \\
   overline{z}_2 z_3).\n\\end{align*}
...
Adding these two equations, we get\n\\[|z_1 - z_2|^2 + |z_1 - z_3|^2 + |
   z_2 - z_3|^2 + |z_1 + z_2 + z_3|^2 = 3|z_1|^2 + 3|z_2|^2 + 3|z_3
   |^2.\\]Therefore,\n\\begin{align*}\n|z_1 - z_2|^2 + |z_1 - z_3|^2 + |
   z_2 - z_3|^2 &= 3|z_1|^2 + 3|z_2|^2 + 3|z_3|^2 - |z_1 + z_2 + z_3|^2
   \\\\\n&\\le 3 \\cdot 2^2 + 3 \\cdot 3^2 + 3 \\cdot 4^2 \\\\\n&= 87.\n
   \\end{align*}For equality to occur, we must have $z_1 + z_2 + z_3 = 0.
   $ Without loss of generality, we can assume that $z_1 = 2.$
...
[asy]\nunitsize(1 cm);\n\npair zone, ztwo, zthree;\n\nzone = (2,0);\nztwo
    = (3/4,3*sqrt(15)/4);\nzthree = (-11/4,-3*sqrt(15)/4);\n\ndraw(
   Circle((0,0),2),red);\ndraw(Circle((0,0),3),green);\ndraw(Circle((0,0)
   ,4),blue);\ndraw(zone--ztwo--zthree--cycle);\n\ndot("$z_1$", zone, E)
   ;\ndot("$z_2$", ztwo, N);\ndot("$z_3$", zthree, SW);\n[/asy]\n\
   nAlternative: For equality to occur, we must have $z_1 + z_2 + z_3 =
   0.$ Without loss of generality, we can assume that $z_1 = 2.$ Then
   $z_2 + z_3 = -2.$ Let $z_2 = x + iy$ so that $z_3 = -x - 2 - iy,$
   where $x$ and $y$ are real numbers. We need\n\\begin{align*}\n |z_2
   |^2 = x^2 + y^2 &= 9 \\\\\n |z_3|^2 = (x + 2)^2 + y^2 &= 16.\n\\end{
   align*}Subtracting the first equation from the second, we get $4x + 4
    = 7,$ or $x = \\dfrac34.$ One solution is $z_2 = \\dfrac34 + i\\
   dfrac{3\\sqrt{15}}{4}$ and $z_3 = -\\dfrac{11}4 + i\\dfrac{3\\sqrt
   {15}}{4}.$ This example shows that equality is possible, so the
   maximum value is $\\boxed{87}.$'
```

Listing 9: Failure case for TORA: **Syntax Error**.

```
Idx: 106
Type: Algebra
Level: Level 5
Question: 'Below is a portion of the graph of a function, $y=h(x)$:\n\n[
   asy]\nimport graph; size(8cm); real lsf=0.5; pen dps=linewidth(0.7)+
   fontsize(10); defaultpen(dps); pen ds=black; real xmin=-0.75,xmax
   =8.25,ymin=-1.25,ymax=10.25;\n\npen cqcqcq=rgb(0.75,0.75,0.75);\n\n/*
   grid*/ pen gs=linewidth(0.7)+cqcqcq+linetype("2 2"); real gx=1,gy=1;\
```

```
     nfor(real i=ceil(xmin/gx)*gx;i<=floor(xmax/gx)*gx;i+=gx) draw((i,ymin)
        --(i,ymax),gs); for(real i=ceil(ymin/gy)*gy;i<=floor(ymax/gy)*gy;i+=
        gy) draw((xmin,i)--(xmax,i),gs);\n\nLabel laxis; laxis.p=fontsize(10)
        ;\n\nxaxis("",xmin,xmax,Ticks(laxis,Step=1.0,Size=2,NoZero),Arrows(6),
        above=true); yaxis("",ymin,ymax,Ticks(laxis,Step=1.0,Size=2,NoZero),
        Arrows(6),above=true);\nreal f1(real x){return (x-0.5)*(x-2.5)*(x-6.5)
        *(x-7.5)/16+x;}\ndraw(graph(f1,-0.25,8.25),linewidth(1));\nclip((xmin,
        ymin)--(xmin,ymax)--(xmax,ymax)--(xmax,ymin)--cycle);\nlabel("$y=h(x)
        $",(8.5,8),E);\n[/asy]\n\nWhat is the sum of all integers $x$ in the
        interval shown ($0\\le x\\le 8$) such that $h(x)>x$ ?'
Solution:
 '''python
from sympy import symbols, solve

def sum_of_x():
    """Below is a portion of the graph of a function, $y=h(x)$:

[asy]
import graph; size(8cm); real lsf=0.5; pen dps=linewidth(0.7)+fontsize
    (10); defaultpen(dps); pen ds=black; real xmin=-0.75,xmax=8.25,ymin
    =-1.25,ymax=10.25;

pen cqcqcq=rgb(0.75,0.75,0.75);

/*grid*/ pen gs=linewidth(0.7)+cqcqcq+linetype("2 2"); real gx=1,gy=1;
for(real i=ceil(xmin/gx)*gx;i<=floor(xmax/gx)*gx;i+=gx) draw((i,ymin)--(i,
    ymax),gs); for(real i=ceil(ymin/gy)*gy;i<=floor(ymax/gy)*gy;i+=gy)
    draw((xmin,i)--(xmax,i),gs);

Label laxis; laxis.p=fontsize(10);

xaxis("",xmin,xmax,Ticks(laxis,Step=1.0,Size=2,NoZero),Arrows(6),above=
    true); yaxis("",ymin,ymax,Ticks(laxis,Step=1.0,Size=2,NoZero),Arrows
    (6),above=true);
real f1(real x){return (x-0.5)*(x-2.5)*(x-6.5)*(x-7.5)/16+x;}
draw(graph(f1,-0.25,8.25),linewidth(1));
clip((xmin,ymin)--(xmin,ymax)--(xmax,ymax)--(xmax,ymin)--cycle);
label("$y=h(x)$",(8.5,8),E);
[/asy]

What is the sum of all integers $x$ in the interval shown ($0\le x\le 8$)
    such that $h(x)>x$ ?"""
    x = symbols('x')
    h_x = (x - 0.5) * (x - 2.5) * (x - 6.5) * (x - 7.5) / 16 + x

    # Find the x-values where h(x) > x
    x_values = [x for x in range(9) if h_x.subs(x, x) > x]

    # Calculate the sum of these x-values
    sum_x = sum(x_values)

    return sum_x

result = sum_of_x()
print(result)
'''
'''output
TypeError: cannot determine truth value of Relational
'''
```

The code above has an error. The `if` statement can only be used with
    boolean values, but the `h_x.subs(x, x) > x` comparison is not a
    boolean value. We need to find a different way to solve this problem.

Let's analyze the problem. We are asked to find the sum of all integers
    $x$ in the interval shown ($0\le x\le 8$) such that $h(x)>x$.

We can see that $h(x)$ is a parabola that opens upwards. For a parabola
    that opens upwards, the maximum value of the parabola will occur at
    the midpoint of the interval, which is $x = \frac{0 + 8}{2} = 4$.

For $x < 4$, $h(x)$ is below the line $y = x$, so $h(x) \le x$. For $x >
    4$, $h(x)$ is above the line $y = x$, so $h(x) > x$.

Therefore, we only need to find the sum of the integers $x$ in the
    interval $0 \le x \le 4$ such that $h(x) > x$.

Here's the corrected code:

```python
from sympy import symbols, solve

def sum_of_x():
    """
    Find the sum of all integers x in the interval shown ($0\le x\le 8$)
        such that h(x)>x.

    Returns:
        int: The sum of all integers x in the interval shown ($0\le x\le 8$
            ) such that h(x)>x.
    """
    x = symbols('x')
    h_x = (x - 0.5) * (x - 2.5) * (x - 6.5) * (x - 7.5) / 16 + x

    # Find the x-values where h(x) > x and 0 <= x <= 4
    x_values = [x for x in range(5) if h_x.subs(x, x) > x]

    # Calculate the sum of these x-values
    sum_x = sum(x_values)

    return sum_x

result = sum_of_x()
print(result)
```
```output
SyntaxError: invalid decimal literal
```
The code above still has an error. The `h_x.subs(x, x) > x` comparison is
    not a boolean value. We need to find a different way to solve this
    problem.

Let's analyze the problem. We are asked to find the sum of all integers
    $x$ in the interval shown ($0\le x\le 8$) such that $h(x)>x$.

We can see that $h(x)$ is a parabola that opens upwards. For a parabola
    that opens upwards, the maximum value of the parabola will occur at
    the midpoint of the interval, which is $x = \frac{0 + 8}{2} = 4$.

For $x < 4$, $h(x)$ is below the line $y = x$, so $h(x) \le x$. For $x >
    4$, $h(x)$ is above the line $y = x$, so $h(x) > x$.

Therefore, we only need to find the sum of the integers $x$ in the
    interval $0 \le x \le 4$ such that $h(x) > x$.

...

Here's the corrected code:

```python
def sum_of_x():
```

```
    """
    Find the sum of all integers x in the interval shown ($0\le x\le 8$)
        such that h(x)>x.

    Returns:
        int: The sum of all integers x in the interval shown ($0\le x\le 8$
            ) such that h(x)>x.
    """
    # Define the interval
    interval = range(5)

    # Find the x-values where h(x) > x
    x_values = [x for x in interval if x * (x - 4) * (x - 6) * (x - 7) /
        16 > x]

    # Calculate the sum of these x-values
    sum_x = sum(x_values)

    return sum_x

result = sum_of_x()
print(result)
```
```output
SyntaxError: invalid decimal literal
```
Reach max function call limit.

Execution: None
Prediction: 'SyntaxError:invaliddecimalliteral'
Ground Truth: '26'
Score: [False]
Reference CoT: 'We could check $h(x)$ separately for each integer $x$
    from $0$ to $8$: for example, $h(0)\\approx 3.8$, so $h(0)>0$, but $h
    (1)\\approx -0.7$, so $h(1)\\not>1$, and so on.\n\nHowever, it is
    easier to see at a glance which $x$ satisfy $h(x)>x$ by superimposing
     a graph of $y=x$ on the graph of $y=h(x)$:\n\n[asy]\ndraw
    ((-0.75,-0.75)--(8.25,8.25),red+1);\nimport graph; size(8cm); real
    lsf=0.5; pen dps=linewidth(0.7)+fontsize(10); defaultpen(dps); pen ds
    =black; real xmin=-0.75,xmax=8.25,ymin=-1.25,ymax=10.25;\n\npen
    cqcqcq=rgb(0.75,0.75,0.75);\n\n/*grid*/ pen gs=linewidth(0.7)+cqcqcq+
    linetype("2 2"); real gx=1,gy=1;\nfor(real i=ceil(xmin/gx)*gx;i<=
    floor(xmax/gx)*gx;i+=gx) draw((i,ymin)--(i,ymax),gs); for(real i=ceil
    (ymin/gy)*gy;i<=floor(ymax/gy)*gy;i+=gy) draw((xmin,i)--(xmax,i),gs);\
    n\nLabel laxis; laxis.p=fontsize(10);\n\nxaxis("",xmin,xmax,Ticks(
    laxis,Step=1.0,Size=2,NoZero),Arrows(6),above=true); yaxis("",ymin,
    ymax,Ticks(laxis,Step=1.0,Size=2,NoZero),Arrows(6),above=true);\nreal
     f1(real x){return (x-0.5)*(x-2.5)*(x-6.5)*(x-7.5)/16+x;}\ndraw(graph
    (f1,-0.25,8.25),linewidth(1));\nclip((xmin,ymin)--(xmin,ymax)--(xmax,
    ymax)--(xmax,ymin)--cycle);\nlabel("$y=h(x)$",(8.5,8),E);\ndot((0,0),
    blue); dot((3,3),blue); dot((4,4),blue); dot((5,5),blue); dot((6,6),
    blue); dot((8,8),blue);\n[/asy]\n\nThe six blue dots above mark the
    integer points $(x,x)$ which lie below the graph of $y=h(x)$,
    indicating that $h(x)>x$. Their $x$-coordinates are $0,3,4,5,6,8,$
    which add up to $\\boxed{26}$.'
```

Listing 10: Failure case for TORA: **Runtime Error**. TORA failed to fall back from program timeout to a rationale-based solution. For Intermediate Algebra and Precalculus problems like this example, direct brute-force solutions are often infeasible, resulting in exceptions like timeout. Addressing these challenges requires complex reasoning over math expressions, and may involve sophisticated problem-solving and proof techniques like forward and backward reasoning (Jiang et al., 2023), as well as result verification (Gou et al., 2023; Zhou et al., 2023a).

```
Idx: 2261
```

```
Yype: Intermediate Algebra
Level: Level 3
Question: 'For some positive integers $m$ and $n,$\n\\[\\sqrt{23 + \\sqrt
    {28}} + \\sqrt{23 - \\sqrt{28}} = \\sqrt{m + \\sqrt{n}}.\\]Enter the
    ordered pair $(m,n).$'
Solution:
```python
from sympy import sqrt, solve, symbols

def find_m_n():
    """For some positive integers $m$ and $n,$
\[\sqrt{23 + \sqrt{28}} + \sqrt{23 - \sqrt{28}} = \sqrt{m + \sqrt{n}}.\]
    Enter the ordered pair $(m,n).$"""
    m, n = symbols('m n')

    # Define the given equation
    equation = sqrt(23 + sqrt(28)) + sqrt(23 - sqrt(28)) - sqrt(m + sqrt(n
        ))

    # Solve the equation for m and n
    solutions = solve(equation, (m, n))

    # Filter out the solutions where m and n are not positive integers
    solutions = [(sol_m, sol_n) for sol_m, sol_n in solutions if sol_m > 0
         and sol_n > 0 and sol_m.is_integer and sol_n.is_integer]

    return solutions[0]

result = find_m_n()
print(result)
```
```output
TypeError: cannot determine truth value of Relational
```
The error in the code above is due to the fact that the `solve` function
    in SymPy does not handle square roots very well. Instead, we can
    manually solve the equation for $m$ and $n$ using the properties of
    square roots.

Let's denote $\sqrt{23 + \sqrt{28}}$ as $a$ and $\sqrt{23 - \sqrt{28}}$
    as $b$. Then the given equation can be rewritten as $a + b = \sqrt{m +
     \sqrt{n}}$.

Squaring both sides, we get $a^2 + 2ab + b^2 = m + \sqrt{n}$.

Since $a^2 = 23 + \sqrt{28}$ and $b^2 = 23 - \sqrt{28}$, we can
    substitute these values into the equation to get $28 + 2\sqrt{28} +
    23 - \sqrt{28} = m + \sqrt{n}$.

Solving this equation for $m$ and $n$ gives us $m = 49$ and $n = 72$.

Therefore, the ordered pair $(m, n)$ is $\boxed{(49, 72)}$.

Execution: None
Prediction: (49,72)
Ground Truth: (46,2004)
Score: [False]
Reference CoT: 'Let $x = \\sqrt{23 + \\sqrt{28}} + \\sqrt{23 - \\sqrt
    {28}}.$ Then\n\\begin{align*}\nx^2 &= 23 + \\sqrt{28} + 2 \\sqrt{23 +
     \\sqrt{28}} \\sqrt{23 - \\sqrt{28}} + 23 - \\sqrt{28} \\\\\n&= 46 +
    2 \\sqrt{23^2 - 28} \\\\\n&= 46 + 2 \\sqrt{501} \\\\\n&= 46 + \\sqrt
    {2004}.\n\\end{align*}Hence, $(m,n) = \\boxed{(46,2004)}.$'
```

Listing 11: Failure case for TORA: **Rationale-only Error**.

```
Idx: 2638
Type: Intermediate Algebra
Level: Level 1
Question: 'The graph of $y = f(x)$ is shown below.\n\n[asy]\nunitsize(0.5
    cm);\n\nreal func(real x) {\n real y;\n if (x >= -3 && x <= 0) {y =
    -2 - x;}\n if (x >= 0 && x <= 2) {y = sqrt(4 - (x - 2)^2) - 2;}\n if
    (x >= 2 && x <= 3) {y = 2*(x - 2);}\n return(y);\n}\n\nint i, n;\n\
    nfor (i = -5; i <= 5; ++i) {\n draw((i,-5)--(i,5),gray(0.7));\n draw
    ((-5,i)--(5,i),gray(0.7));\n}\n\ndraw((-5,0)--(5,0),Arrows(6));\ndraw
    ((0,-5)--(0,5),Arrows(6));\n\nlabel("$x$", (5,0), E);\nlabel("$y$",
    (0,5), N);\n\ndraw(graph(func,-3,3),red);\n\nlabel("$y = f(x)$",
    (3,-2), UnFill);\n[/asy]\n\nWhich is the graph of $y = -f(x)$?\n\n[
    asy]\nunitsize(0.5 cm);\n\npicture[] graf;\nint i, n;\n\nreal func(
    real x) {\n real y;\n if (x >= -3 && x <= 0) {y = -2 - x;}\n if (x >=
     0 && x <= 2) {y = sqrt(4 - (x - 2)^2) - 2;}\n if (x >= 2 && x <= 3) {
    y = 2*(x - 2);}\n return(y);\n}\n\nreal funcb(real x) {\n return(-
    func(x));\n}\n\nreal funcd(real x) {\n return(-func(-x));\n}\n\nreal
    funce(real x) {\n return(func(-x));\n}\n\nfor (n = 1; n <= 5; ++n) {\
    n graf[n] = new picture;\n for (i = -5; i <= 5; ++i) {\n draw(graf[n
    ],(i,-5)--(i,5),gray(0.7));\n draw(graf[n],(-5,i)--(5,i),gray(0.7));\
    n }\n draw(graf[n],(-5,0)--(5,0),Arrows(6));\n draw(graf[n],(0,-5)
    --(0,5),Arrows(6));\n\n label(graf[n],"$x$", (5,0), E);\n label(graf[
    n],"$y$", (0,5), N);\n}\n\ndraw(graf[1],(-5,3)--(-2,0),red);\ndraw(
    graf[1],arc((-2,2),2,270,360),red);\ndraw(graf[1],(0,2)--(2,4),red);\
    ndraw(graf[2],graph(funcb,-3,3),red);\ndraw(graf[3],(-3,2)--(-2,0),
    red);\ndraw(graf[3],arc((-2,-2),2,0,90),red);\ndraw(graf[3],(0,-2)
    --(3,-5),red);\ndraw(graf[4],graph(funcd,-3,3),red);\ndraw(graf[5],
    graph(funce,-3,3),red);\n\nlabel(graf[1], "A", (0,-6));\nlabel(graf
    [2], "B", (0,-6));\nlabel(graf[3], "C", (0,-6));\nlabel(graf[4], "D",
     (0,-6));\nlabel(graf[5], "E", (0,-6));\n\nadd(graf[1]);\nadd(shift
    ((12,0))*(graf[2]));\nadd(shift((24,0))*(graf[3]));\nadd(shift((6,-12)
    )*(graf[4]));\nadd(shift((18,-12))*(graf[5]));\n[/asy]\n\nEnter the
    letter of the graph of $y = -f(x).$'
Solution:
The correct graph is the one that reflects the original graph y = f(x)
    about the y-axis. This is graph D.

So, the answer is $\boxed{\text{D}}$.

Execution: None
Prediction: 'D'
Ground Truth: 'B'
Score: [False]
Reference CoT: 'The graph of $y = -f(x)$ is the reflection of the graph
    of $y = f(x)$ in the $x$-axis. The correct graph is $\\boxed{\\text{B
    }}.$'
```

Listing 12: Failure case for TORA: **False Negative**.

```
Idx: 2228
Type: Intermediate Algebra
Level: Level 3
Question: 'Find all the real roots of\n\\[(12x - 1)(6x - 1)(4x - 1)(3x -
    1) = 5.\\]Enter all the real roots, separated by commas.'
Solution:
 ```python
from sympy import symbols, solve, simplify

def find_real_roots():
    """Find all the real roots of
\[(12x - 1)(6x - 1)(4x - 1)(3x - 1) = 5.\]Enter all the real roots,
    separated by commas."""
    x = symbols('x')
    equation = (12*x - 1) * (6*x - 1) * (4*x - 1) * (3*x - 1) - 5
```

```
    roots = solve(equation, x)
    real_roots = [root for root in roots if root.is_real]

    return real_roots

result = find_real_roots()
print(result)
```
```output
[-1/12, 1/2]
```
The real roots of the equation are $\boxed{-\frac{1}{12}, \frac{1}{2}}$.

Execution: None
Prediction: '$-\frac{1}{12}, \frac{1}{2}$' Ground Truth: '$\frac{1}{2}, -\frac{1}{12}$' Score: [False]
Reference CoT: 'We can expand using the pairs $(3x - 1)(12x - 1) = 36x^2 - 15x + 1$ and $(6x - 1)(4x - 1) = 24x^2 - 10x + 1,$ so\n\\[(36x^2 - 15x + 1)(24x^2 - 10x + 1) = 5.\\]Let $y = 12x^2 - 5x.$ Then\n\\[(3y + 1)(2y + 1) = 5.\\]This simplifies to $6y^2 + 5y - 4 = 0,$ which factors as $(2y - 1)(3y + 4) = 0.$ Hence, $y = \\frac{1}{2}$ or $y = -\\frac{4}{3}.$\n\nIf $12x^2 - 5x = \\frac{1}{2},$ then $24x^2 - 10x - 1 = 0,$ which factors as\n\\[(2x - 1)(12x + 1) = 0.\\]Hence, $x = \\frac{1}{2}$ or $x = -\\frac{1}{12}.$\n\nIf $12x^2 - 5x = -\\frac{4}{3},$ then\n\\[36x^2 - 15x + 4 = 0,\\]which has no real solutions.\n\nTherefore, the real roots are $\\boxed{\\frac{1}{2}, -\\frac{1}{12}}.$'
```

