# OpenReview forum: "ToRA: A Tool-Integrated Reasoning Agent for Mathematical Problem Solving"
_ICLR.cc/2024/Conference — ICLR 2024 poster_

### Official Review · Reviewer_Gq26 · 2023-10-21

**Soundness:** 3 good
**Presentation:** 3 good
**Contribution:** 2 fair
**Rating:** 6
**Confidence:** 3

**Summary:**

The paper presents ToRA, a method of utilising interleaved reasoning and tool-use traces to improve language models' capability in mathematics. The authors used GPT-4 to get a collection of interactive tool-use trajectories for mathematical problem solving, and fine-tuned open-sourced language models on this collection to improve their performance.

**Strengths:**

- The performance gains on the downstream mathematical reasoning benchmarks is impressive.
- The ablation studies convincingly proved the necessity of both the rationale and the program parts of the ToRA pipeline.

**Weaknesses:**

- Adopting complicated mathematical fonts when unnecessary only takes from the readability of the paper. There is no need to use them when the default fonts suffice.

**Questions:**

How should we improve upon ToRA? Since the pipeline relies on a proprietary model to generate the dataset, and the final performance of the strongest ToRA is still inferior to GPT-4, how can we further improve?

---

> ### Author Response · Authors · 2023-11-17
> **Author Response to Official Review by Reviewer Gq26**
>
> Dear reviewer Gq26,
>
> We appreciate your thorough review and insightful comments on our paper. We are pleased that the you recognizes our efforts and the promising results of the ToRA pipeline. We will address the concerns raised and answer the questions posed to improve our work further.
>
> ### **Enhancing Readability with Appropriate Mathematical Fonts**
>
> > Adopting complicated mathematical fonts when unnecessary only takes from the readability of the paper. There is no need to use them when the default fonts suffice.
> >
>
> We appreciate your feedback! We understand the reviewer's concern about the complexity of mathematical fonts making the paper less readable. Our intention was to maintain the mathematical accuracy and precision, e.g., our choice to use $\wp$ for "prompt" and $p$ for "program" was driven by a desire to avoid confusion, but we understand that this might have inadvertently complicated the presentation. To enhance readability, we will revise these symbols to more commonly used and easily comprehensible ones without losing the distinction between the two terms.
>
> ### **Suggestions for Further Improvements on Math Reasoning**
>
> > How should we improve upon ToRA? Since the pipeline relies on a proprietary model to generate the dataset, and the final performance of the strongest ToRA is still inferior to GPT-4
> >
>
> Thank you for posing an insightful question about improving upon ToRA! One point that needs clarification is that the core method of ToRA is tool-integrated reasoning learning, and the data annotation in this process is **model-independent** for collecting tool-integrated reasoning trajectories. It can also be obtained through other open-source models or even manually annotated for higher-quality data.
>
> Moreover, our experiments substantiated the effectiveness of this tool-integrated reasoning format on GPT-4, demonstrating its potential to further augment GPT-4's performance. As illustrated in Fig. 4 (right), the implementation of tool-integrated reasoning can yield a 9.8% improvement compared to PAL.
>
> Looking forward, we see several main directions to further enhance mathematical reasoning based on ToRA:
>
> - **Continue Pre-training**: We found in subsequent experiments that applying ToRA to models continue pre-trained on math and code data brings a more dramatic improvement! This suggests that pre-training and data engineering might deserve more attention for the research community!
> - **SFT Mixture Scale-up**: We could explore amplifying the quality and coverage of fine-tuning data in tool-integrated reasoning format, which might further improve generalization and robustness.
> - **Reward Modeling**: We could explore methods to enhance tool-integrated reasoning by incorporating reward modeling methods like [1], to further improve accuracy, intention alignment, interpretability, and to make it more user-friendly.
>
> We extend our gratitude for your constructive feedback and the opportunity to clarify the nuances of our work. We believe that the proposed changes will significantly improve our paper's readability and impact.
>
> [1] Lightman, Hunter, et al. "Let's Verify Step by Step." arXiv preprint arXiv:2305.20050 (2023).

---

> > ### Comment · Reviewer_Gq26 · 2023-11-19
> >
> > Thank you for your reply! The suggested improvements are very promising. I look forward to seeing them.
> >
> > I shall retain my rating.

---

> > > ### Author Response · Authors · 2023-11-20
> > > **Author Follow-Up to Reviewer Gq26**
> > >
> > > Thank you for your continued interest in our research! We're excited to share our preliminary results from applying ToRA to models that are continue pre-trained (domain adaptive pre-training, DAPT [1]) on math and code corpus [2,3]. This has resulted in "ToRA-Code + DAPT”. As demonstrated in the table below, this approach has yielded an average improvement of 4.6% across all datasets, surpassing the previous state-of-the-art 7B model, ToRA-Code-7B.
> > >
> > > Model         |Size| GSM8k| MATH| GSM-Hard | SVAMP | TabMWP | ASDiv | MAWPS | AVG
> > > --------------|----|------|-----|------|-------|--------|-------|-------|-----
> > > ToRA-Code     |7B  | 72.6 | 44.6| 56.0 | 70.4  | 51.6   | 78.7  | 91.3  | 66.5
> > > ToRA-Code + DAPT|7B  | **74.8** | **49.5** | **59.0** | **76.0**  | **63.5**   | **82.3**  | **92.8**  | **71.1**
> > >
> > > We’ll will disclose more results in our future work. We sincerely value your review and guidance in improving our paper!
> > >
> > > ### References
> > >
> > > [1] Gururangan, Suchin, et al. "Don't stop pretraining: Adapt language models to domains and tasks." *arXiv preprint arXiv:2004.10964* (2020).
> > >
> > > [2] https://huggingface.co/datasets/open-web-math/open-web-math
> > >
> > > [2] Azerbayev, Zhangir, et al. "Llemma: An open language model for mathematics." arXiv preprint arXiv:2310.10631 (2023).

---

### Official Review · Reviewer_U6Sz · 2023-10-30

**Soundness:** 2 fair
**Presentation:** 2 fair
**Contribution:** 2 fair
**Rating:** 5
**Confidence:** 5

**Summary:**

This paper introduces TORA,
 a series of Tool-integrated Reasoning Agents designed to enhance mathematical problem-solving by combining natural language reasoning with external tools. TORA models are trained using interactive tool-use trajectories, employing imitation learning and output space shaping techniques. Experimental results demonstrate that TORA outperforms open-source models on various mathematical reasoning datasets.

**Strengths:**

- The paper is easy to follow
- TORA achieves good performance on math datasets

**Weaknesses:**

- **Limited of technical novelty**:
  - Using imitation learning to improve the mathematical reasoning ability of open-source models has been proposed in many recent works, e.g.,
    - Scaling relationship on learning mathematical reasoning with large language models, https://arxiv.org/abs/2308.01825
    - WizardMath: Empowering Mathematical Reasoning for Large Language Models via Reinforced Evol-Instruct, https://arxiv.org/abs/2308.09583
    - MetaMath: Bootstrap Your Own Mathematical Questions for Large Language Models, https://arxiv.org/abs/2309.12284
    - MAmmoTH: Building Math Generalist Models through Hybrid Instruction Tuning, https://arxiv.org/abs/2309.05653
  - At the Output Space Shaping step, the authors use nucleus sampling to generate more reasoning paths and pick the corrected paths. This is a technique widely used in the works listed above. The only difference is that this paper fixes some of the preceding portions of wrong trajectories, while existing works resample the whole trajectory. However, validating which portions of trajectories are correct is very challenging, and the authors need to **enumerate** possible preceding portions of wrong trajectories, which is time-consuming. In my opinion, the existing method (i.e., re-sampling and picking the correct paths) is simpler and may be more effective.

- **Concerns on reproducibility**: As training data are not provided, reviewers and readers can't check the reproducibility of this paper. Note that existing works like WizardMath, MetaMath, and MAmmoTH have released their training data for the community to reproduce their results. Moreover, checkpoints of TORA are not provided in the appendix for checking reproducibility.



- "TORA outperforms WizardMath by around 45% in Algebra and Number Theory, **which is attributed to stimulating and shaping tool-use behavior**." From Table 3, we cannot conclude that the better performance of TORA is due to stimulating and shaping tool-use behavior, as
WizardMath uses augmented data **from LLaMA**, while TORA uses data generated **from GPT-4**. Note that GPT-4 is much more powerful than LLaMA.

- In section 2.2, greedy decoding is used for generating trajectories from GPT-4. Thus, only one path per question can be obtained in TORA-CORPUS dataset. In my opinion, the accuracy of LLaMA trained on TORA-CORPUS has yet to saturate (e.g., plot the accuracy w.r.t. #samples of TORA-CORPUS).  To generate more trajectories, a simple approach (which is widely used in the above works) is to use temperature sampling (rather than greedy decoding)  and pick the correct ones for training.

- As temperature sampling can generate more samples from GPT-4, TORA-CORPUS can be more diverse (compared with greedy decoding). To verify the effectiveness of Output Space Shaping in Section 3.5.2, it is better to **augment more data from GPT-4 and let the accuracy of LLaMA trained on the TORA-CORPUS saturates first**. Otherwise, it is difficult to say whether the improvement of LLaMA is from more training data or the proposed Output Space Shaping.

- Ablation study of hyperparameter $n$ (maximum rounds). in experiments, $n=3$ is used. Question: is the performance of TORA sensitive to $n$?

- writing:
  - "forward and backward reasoning, as well as result verification": references for forward/backward reasoning, verification
  - "Addressing these challenges requires complex symbolic reasoning over algebraic expressions": references for symbolic reasoning
  - where is the definition of $\theta$ in (4)?

**Questions:**

- In the Conclusion section, the authors mention that "our systematic analysis ... paving the way for the development of more advanced and versatile reasoning agents". **How and why does this analysis pave the way?**
- What is the difference between Tool-Integrated Reasoning (Algorithm 1) and existing methods (e.g., PAL (Gao et al., 2022), PoT (Program of Thoughts Prompting: Disentangling Computation from Reasoning for Numerical Reasoning Tasks, https://arxiv.org/abs/2211.12588))?
- "For numerical values, we perform **rounding**, while for expressions, we employ **sympy** for parsing." Are these two techniques used in the baseline methods?

- Are there any empirical results can support the ``Output Space Shaping improves **diversity**''. Furthermore, the diversity measure is not defined in the paper.

---

> ### Author Response · Authors · 2023-11-17
> **Author Response to Official Review by Reviewer U6Sz (1/3)**
>
> Dear Reviewer U6Sz,
>
> Thank you for your time and thoughtful feedback on ToRA! We have addressed each of your points below.
>
> ### **Technical Novelty**
>
> > Using imitation learning to improve the mathematical reasoning ability of open-source models has been proposed in many recent works, e.g.,
> >
> > - Scaling relationship on learning mathematical reasoning with large language models, https://arxiv.org/abs/2308.01825
> > - WizardMath: Empowering Mathematical Reasoning for Large Language Models via Reinforced Evol-Instruct, https://arxiv.org/abs/2308.09583
> > - MetaMath: Bootstrap Your Own Mathematical Questions for Large Language Models, https://arxiv.org/abs/2309.12284
> > - MAmmoTH: Building Math Generalist Models through Hybrid Instruction Tuning, https://arxiv.org/abs/2309.05653
>
> We appreciate your reference to concurrent works. While both ToRA and these studies employ imitation learning, the Tool-integrated Reasoning format proposed to train ToRA fundamentally differs from the CoT or PoT formats used in these works, yielding significantly better results.
>
> Specifically, references [1-3] primarily adopt natural language rationale (CoT), focusing on augmenting CoT solutions [1] and questions [2, 3] for improvements. On the other hand, reference [4] concentrates on hybrid training using CoT and PoT.
>
> In contrast, ToRA introduces the Tool-integrated reasoning format, which brings clear advantages. For instance, even the smallest ToRA-Code-7B outperforms MetaMath-70B [3] (44.6% vs. 26.0%) and MAmmoTH-Coder-34B [4] (44.6% vs. 43.6%).
>
> ### **Output Space Shaping**
>
> > At the Output Space Shaping step, the authors use nucleus sampling to generate more reasoning paths and pick the corrected paths. This is a technique widely used in the works listed above. The only difference is that this paper fixes some of the preceding portions of wrong trajectories, while existing works resample the whole trajectory. However, validating which portions of trajectories are correct is very challenging, and the authors need to enumerate possible preceding portions of wrong trajectories, which is time-consuming.
> >
>
> We proposed the method of **learning from small model errors corrected by a teacher model**, which we call **teacher correction**, and we combined it with sampling for output space shaping.
>
> The step-level enumeration for teacher correction is a one-time cost for constructing training data. In our experiments, compared with using sampling alone for output space shaping, introducting teacher correction does not increase training cost because we maintain the same volume of training data.
>
> > In my opinion, the existing method (i.e., re-sampling and picking the correct paths) is simpler and may be more effective.
> >
>
> Our research shows that integrating teacher correction contributes to higher efficacy than merely using sampling. As illustrated in Figure 5, employing the sampling strategy alone resulted in a 2.7% performance enhancement. Conversely, the incorporation of the correction strategy led to a performance improvement ranging from 3.4% to 4.0% across two datasets. This improvement was achieved while maintaining the same volume of training data as with sampling alone, thereby substantiating the effectiveness of our output space shaping strategies.
>
> ### **Reproducibility**
>
> > Concerns on reproducibility: As training data are not provided, reviewers and readers can't check the reproducibility of this paper. Note that existing works like WizardMath, MetaMath, and MAmmoTH have released their training data for the community to reproduce their results. Moreover, checkpoints of TORA are not provided in the appendix for checking reproducibility.
> >
>
> We understand your concerns about reproducibility. We're currently undergoing an internal review to open-source the ToRA-Corpus and will add the open-source links to code, model checkpoints, etc., after the anonymous review period.

---

> > ### Author Response · Authors · 2023-11-17
> > **Author Response to Official Review by Reviewer U6Sz (2/3)**
> >
> > ### **Performance Attribution**
> >
> > > "TORA outperforms WizardMath by around 45% in Algebra and Number Theory, which is attributed to stimulating and shaping tool-use behavior." From Table 3, we cannot conclude that the better performance of TORA is due to stimulating and shaping tool-use behavior, as WizardMath uses augmented data from LLaMA, while TORA uses data generated from GPT-4. Note that GPT-4 is much more powerful than LLaMA.
> > >
> >
> > We appreciate your comments and would like to address the points raised regarding the performance of ToRA and WizardMath.
> >
> > In the WizardMath paper, it is mentioned that they "for each instruction, … use ChatGPT and Wizard-E to generate 2~4 evolved instructions" and "depend on ChatGPT to provide process supervision" [2]. This suggests that WizardMath uses ChatGPT (not solely LLaMA) for data augmentation, as well as the human expert annotations within the original datasets (>96k in total). On the other hand, ToRA only uses GPT-4 annotated 16k data and applies output space shaping, and yet it achieves much higher performance (e.g., 44.6% vs. 10.7% on MATH). We would like to elaborate on why we attribute these gains predominantly to stimulating and shaping tool-use behavior.
> >
> > - **Impact of tool-integrated reasoning in a fair setting**: Training LLaMA-2 7B on 16k annotations in Tool-Integrated Reasoning format (i.e., the ToRA-Corpus) is significantly better than using the 15k CoT annotations (33.6% vs. 7.2% in Fig 4), despite the fact that the CoT annotations come from human experts (i.e., the original annotations provided by the datasets) while ToRA Corpus is annotated by GPT-4 with an imperfect annotation accuracy of 83.1%. This demonstrates the effectiveness of tool-use behavior stimulation.
> > - **Additional augmented data used by WizardMath**: Even though WizardMath enhances human-annotated data with ChatGPT augmentation (>96k data in total), it still falls significantly behind using only the ToRA-Corpus (16k data in total) for training (33.6% vs. 10.7%, both base models are LLaMA-2 7B). We argue that this performance discrepancy isn't due to ChatGPT's inferiority to GPT-4, but rather the reasoning format.
> > - **Effect of shaping**: Additionally, our approach to shaping tool-use behavior has indeed resulted in substantial improvement. As Table 3 illustrates, this method boosted the performance of the 7B, 13B, and 34B models by 4.4%, 3.5%, and 3.4%, respectively on the competition-level MATH dataset. For instance, ToRA-34B rose from 47.4% to 50.8%. This enhancement was particularly notable in subtopics such as Precalculus, Geometry, and Algebra, with increases ranging from 5.9% to 6.3%.
> >
> > ### **Generation of Trajectories**
> >
> > > In section 2.2, greedy decoding is used for generating trajectories from GPT-4. Thus, only one path per question can be obtained in TORA-CORPUS dataset. In my opinion, the accuracy of LLaMA trained on TORA-CORPUS has yet to saturate (e.g., plot the accuracy w.r.t. #samples of TORA-CORPUS). To generate more trajectories, a simple approach (which is widely used in the above works) is to use temperature sampling (rather than greedy decoding) and pick the correct ones for training.
> > As temperature sampling can generate more samples from GPT-4, TORA-CORPUS can be more diverse (compared with greedy decoding). To verify the effectiveness of Output Space Shaping in Section 3.5.2, it is better to augment more data from GPT-4 and let the accuracy of LLaMA trained on the TORA-CORPUS saturates first. Otherwise, it is difficult to say whether the improvement of LLaMA is from more training data or the proposed Output Space Shaping.
> > >
> >
> > We appreciate your suggestion. In the original ToRA paper, we have already adopted the sampling method you mentioned to ensure more valid trajectories for each question.
> >
> > **Kindly note**: In fact, right after the sentence you quoted, we stated, "For questions where GPT-4 fails with greedy decoding, we apply nucleus sampling with a sample size of 10 and keep up to four valid trajectories per question.”
> >
> > Furthermore, Output Space Shaping is an algorithmic process that leverages sampling to introduce more diverse reasoning trajectories and introduce teacher correction to reduce reasoning errors. We believe the model used for sampling and the teacher for correction are not restricted to LLaMA models (despite that we used LLaMA in our experiments), but can also be GPT-4.
> >
> > - We also note that there are follow-up works to ToRA that have replaced the teacher with GPT-4 and achieved good results [5].

---

> ### Author Response · Authors · 2023-11-17
> **Author Response to Official Review by Reviewer U6Sz (3/3)**
>
> ### **Setting Maximum Rounds of Interaction**
>
> > Ablation study of hyperparameter $n$ (maximum rounds). in experiments, $n=3$ is used. Question: is the performance of TORA sensitive to $n$?
> >
>
> Our preliminary experiments found that on mathematical reasoning benchmarks like GSM8k and MATH, large language models nearly always solve problems with n ≤ 3 interactions, so we set it as the default hyperparameter. Setting a higher number of interactions does not bring significant gains.
>
> ### **Writing**
>
> > - "forward and backward reasoning, as well as result verification": references for forward/backward reasoning, verification"
> > - Addressing these challenges requires complex symbolic reasoning over algebraic expressions": references for symbolic reasoning
> > - where is the definition of $\theta$ in (4)?
>
> We appreciate your meticulous attention to details and pointing out writing typos. We will add all the references you mentioned and the definition of $\theta$ in equation (4) in future updates.
>
> ### **Questions**
>
> > In the Conclusion section, the authors mention that "our systematic analysis ... paving the way for the development of more advanced and versatile reasoning agents". How and why does this analysis pave the way?
> >
>
> We appreciate your question! As an initial exploration of constructing math reasoning agents based on tool-integrated reasoning, we demonstrated the effectiveness of combining natural language with programming language to solve challenging reasoning problems. We believe this approach has potential and is worth further exploration. Additionally, in Section 3.6, we have conducted an in-depth analysis of the areas that still require enhancement and merit additional attention. We believe that this systematic analysis and identification of areas for improvement lays the groundwork for the development of more advanced and versatile reasoning agents.
>
> > What is the difference between Tool-Integrated Reasoning (Algorithm 1) and existing methods (e.g., PAL, PoT)?
> >
>
> As shown in Fig. 2, PAL and PoT are Program-based methods to solve tasks with program synthesis, while our proposed Tool-integrated Reasoning format integrates natural language reasoning with program-based tool use, in order to combine the benefits of both worlds.
>
> As shown in Fig 4, the proposed Tool-Integrated Reasoning consistently surpasses Rationale-only and Program-only approaches. Remarkably, Using LLaMA-2, Tool-Integrated Reasoning achieves substantial improvements of 29.0% and 6.7% over Rationale-only and Program-only, respectively. With the closed-source GPT-4, the improvements are 19.1% and 9.8%, respectively. This emphasizes the effectiveness of integrating natural language rationales with programs.
>
> > "For numerical values, we perform rounding, while for expressions, we employ sympy for parsing." Are these two techniques used in the baseline methods?
> >
>
> Sure! For a fair comparison, we adopted the same evaluation for all methods.
>
> > Are there any empirical results can support the ``Output Space Shaping improves diversity''. Furthermore, the diversity measure is not defined in the paper.
> >
>
> Thanks for your question! We constructed many trajectories for each problem through sampling and teacher correction, where the reasoning processes are different but the answers are correct. In other words, improving diversity means increasing the number of different solution paths for the same problem.
>
> ### **References**
>
> [1] Yuan, Zheng, et al. "Scaling relationship on learning mathematical reasoning with large language models." arXiv preprint arXiv:2308.01825 (2023).
>
> [2] Luo, Haipeng, et al. "Wizardmath: Empowering mathematical reasoning for large language models via reinforced evol-instruct." arXiv preprint arXiv:2308.09583 (2023).
>
> [3] Yu, Longhui, et al. "Metamath: Bootstrap your own mathematical questions for large language models." arXiv preprint arXiv:2309.12284 (2023).
>
> [4] Yue, Xiang, et al. "Mammoth: Building math generalist models through hybrid instruction tuning." arXiv preprint arXiv:2309.05653 (2023).
>
> [5] An, Shengnan, et al. "Learning From Mistakes Makes LLM Better Reasoner." arXiv preprint arXiv:2310.20689 (2023).

---

> ### Comment · Reviewer_U6Sz · 2023-11-20
> **Reply to rebuttal**
>
> Thanks for the response.
> Some of my concerns were addressed, but some remain.
>
> - limited novelty: Existing works have shown the effectiveness of imitation learning in improving the performance of open-source models, while this work just shows that PoT or CoT can be replaced with more powerful PoT+CoT (which is also used in MAmmoTH) to boost performance. Moreover, what are the clear advantages in the reply "In contrast, ToRA introduces the Tool-integrated reasoning format, which brings clear advantages"?
> - the accuracy of LLaMA trained on TORA-CORPUS has yet to saturate (e.g., plot the accuracy w.r.t. #samples of TORA-CORPUS). A follow-up question is: Are data generated from Output Space Shaping more effective than data generated by GPT-4? If not, why not just enlarge the latter? I understand GPT-4 is costly, but how do we maintain a good trade-off?
> - "Our systematic analysis ... paving the way for the development of more advanced and versatile reasoning agents" in the conclusion section might be overclaimed. Just a minor comment.
> - MAmmoTH and MetaMath are **missing** the main table (i.e., Table 2).
>
> > For a fair comparison, we adopted the same evaluation for all methods
>
> It seems that some of the results in the main table are copied from their publications,
> I checked some of them but failed to find these techniques (round, `sympy`), e.g., WizardMath (https://arxiv.org/pdf/2308.09583.pdf), RFT (https://arxiv.org/abs/2308.01825)
>
> > Improving diversity means increasing the number of different solution paths for the same problem
>
> Note that different solution paths do not mean they are diverse.
>
> btw, is the current paper the latest? (updating the paper is allowed in ICLR)

---

> > ### Author Response · Authors · 2023-11-21
> > **Author Follow-up to Reviewer U6Sz**
> >
> > Dear reviewer U6Sz,
> >
> > Thank you for your follow-up questions.
> >
> > ### Novelty and Effectiveness Compared to Concurrent Works
> >
> > > - limited novelty: Existing works have shown the effectiveness of imitation learning in improving the performance of open-source models, while this work just shows that PoT or CoT can be replaced with more powerful PoT+CoT (which is also used in MAmmoTH) to boost performance. For example, what are the clear advantages in the reply "In contrast, ToRA introduces the Tool-integrated reasoning format, which brings clear advantages"?
> > >
> > > - MAmmoTH and MetaMath are missing the main table (i.e., Table 2).
> >
> > We clearly stated in our initial response that these two works were concurrent and distinctly different from ToRA. Our first submission to ICLR’24 was on [15 Sept](https://openreview.net/forum?id=Ep0TtjVoap), while the first submissions of [MAmmoTH](https://openreview.net/forum?id=yLClGs770I) and [MetaMath](https://openreview.net/forum?id=N8N0hgNDRt) to ArXiv fell between 11 ~ 22 Sept.
> >
> > ### Use GPT-4 as Teacher Instead
> >
> > > the accuracy of LLaMA trained on TORA-CORPUS has yet to saturate (e.g., plot the accuracy w.r.t. #samples of TORA-CORPUS). A follow-up question is: Are data generated from Output Space Shaping more effective than data generated by GPT-4? If not, why not just enlarge the latter? I understand GPT-4 is costly, but how do we maintain a good trade-off?
> > >
> >
> > As we mentioned in our previous response, we have indeed sampled solutions using GPT-4 (10 per questions for those unsolved with greedy), and we also believe if we sampling more solutions may provide further boost, but this is hard to scale-up due to the expensive cost.
> >
> > Secondly, in our proposed Output Space Shaping, the model used for sampling and the teacher for correction are not restricted to LLaMA models (despite that we used LLaMA in our experiments). They could also be GPT-4, which we believe could provide better performance. For further insights, we refer you to follow-up works that have replaced the teacher with GPT-4 and achieved good results [3].
> >
> > ### Revise Conclusion
> >
> > > "Our systematic analysis ... paving the way for the development of more advanced and versatile reasoning agents" in the conclusion section might be overclaimed. Just a minor comment.
> > >
> >
> > We appreciate your feedback on the conclusion section. We agree that our phrasing could be revised to better reflect the contribution  of our work to the understanding of how integrating natural language reasoning with programming language-based tool usage can enhance mathematical problem-solving.
> >
> > ### Grading
> >
> > > It seems that some of the results in the main table are copied from their publications, I checked some of them but failed to find these techniques (round, sympy), e.g., WizardMath (https://arxiv.org/pdf/2308.09583.pdf), RFT (https://arxiv.org/abs/2308.01825)
> > >
> >
> > We apologize for any confusion. For the GSM-Hard, SVAMP, TabMWP, ASDiv, and MAWPS datasets, all results were obtained by our own reproduction using the same evaluation methods. As for the GSM8k and MATH datasets, we have used the baseline results reported in the original papers to maintain consistency with the prior work.
> >
> > **Kindly Note**: It's worth noting that rounding and parsing are common grading methods in previous works [4], and rounding is actually used in the open-source code of WizardMath and RFT [5, 6].
> >
> > In line with your suggestion, we will re-evaluate and report the results of WizardMath and RFT on GSM8k and MATH using our grader in future updates. These changes will not alter our conclusions. Although different grading methods may lead to slight variations in results, our choice of grading method, similar to [4], is aimed at ensuring more accurate evaluations. Regardless of the grading method used, our conclusions remain unchanged.
> >
> > We deem it crucial to uphold grading fairness and ensure result reproducibility by open-sourcing our evaluation code, which will be linked in the paper post the anonymous review period.
> >
> >
> > ### The definition of “diverse”
> >
> > > Note that different solution paths do not mean they are diverse.
> > >
> >
> > We define diversity as the number of distinct valid reasoning trajectories for a given problem, in line with definitions widely used in previous works like Self-Consistency [7] and DIVERSE [8].  We believe that increasing the number of different solution paths inherently introduces more diversity in reasoning processes, thereby enriching the model's understanding of problem-solving strategies. We are open to suggestions for a more sophisticated measure of diversity if you have any better definition!
> >
> > We have previously updated our paper based on reviewers’ constructive feedback and will continue to make updates as necessary.

---

> > > ### Author Response · Authors · 2023-11-21
> > > **References**
> > >
> > > ### References
> > > [1] Yue, Xiang, et al. "Mammoth: Building math generalist models through hybrid instruction tuning." arXiv preprint arXiv:2309.05653 (2023). https://openreview.net/forum?id=yLClGs770I
> > >
> > > [2] Yu, Longhui, et al. "Metamath: Bootstrap your own mathematical questions for large language models." arXiv preprint arXiv:2309.12284 (2023). https://openreview.net/forum?id=N8N0hgNDRt
> > >
> > > [3]  An, Shengnan, et al. "Learning From Mistakes Makes LLM Better Reasoner." arXiv preprint arXiv:2310.20689 (2023).
> > >
> > > [4] https://github.com/openai/prm800k/blob/main/prm800k/grading/grader.py
> > >
> > > [5] https://github.com/nlpxucan/WizardLM/blob/e7f9b708e41e33c6244ec628c91277e30312181e/WizardMath/inference/gsm8k_inference.py#L36-L47
> > >
> > > [6] https://github.com/OFA-Sys/gsm8k-ScRel/blob/b896be88a0131ea6cc797133893bea6ebd4717d3/eval.py#L81C13-L81C13
> > >
> > > [7] Wang, Xuezhi, et al. "Self-consistency improves chain of thought reasoning in language models." arXiv preprint arXiv:2203.11171 (2022).
> > >
> > > [8] Li, Yifei, et al. "Making language models better reasoners with step-aware verifier." Proceedings of the 61st Annual Meeting of the Association for Computational Linguistics (Volume 1: Long Papers). 2023.

---

> > > ### Author Response · Authors · 2023-11-21
> > > **Author Follow-up to Reviewer U6Sz (2/2)**
> > >
> > > ### The Advantage of Tool-integrated Reasoning
> > >
> > > > What are the clear advantages in the reply "In contrast, ToRA introduces the Tool-integrated reasoning format, which brings clear advantages"?
> > > >
> > >
> > > Solving a complex mathematical problem involves semantic analysis, calculations, answer finalization, etc. Natural language reasoning handles semantic analysis and answer finalization well, while programming excels at complex computations. Hence, we offer a tool-integrated reasoning approach which interleaves natural language and programming to solve a problem. In contrast, MAmmoTH produces either program-only (PoT) or rationale-only (CoT) solutions, and didn’t study the synergy of both in solving a problem.
> > >
> > > Our tool-integrated reasoning format demonstrated significant performance advantages. Figure 4 shows that our approach consistently outperforms Rationale-only and Program-only approaches by 29.0% and 6.7% respectively. Even the smallest ToRA-Code-7B outperforms MetaMath-70B (44.6% vs. 26.0%) and MAmmoTH-Coder-34B (44.6% vs. 43.6%). These results validate the effectiveness of our proposed tool-integrated reasoning format.
> > >
> > > ### Re-evaluate WizardMath Using Our Evaluation Code
> > >
> > > We have re-evaluated WizardMath on GSM8k and MATH using our grading code. And since the best checkpoints of RFT are not open-sourced, we can't reproduce them with our grader. The results are shown below, with rows labeled as "ours" representing our reevaluated results.
> > >
> > > | Model | Size | GSM8k | MATH |
> > > | --- | --- | --- | --- |
> > > | WizardMath | 7B | **54.9** | 10.7 |
> > > | WizardMath (ours) | 7B | 54.7 | **11.2** |
> > > | ToRA | 7B | `68.8` | `40.1` |
> > > |  |  |  |  |
> > > | WizardMath | 13B | **63.9** | 14.0 |
> > > | WizardMath (ours) | 13B | 63.8 | **15.0** |
> > > | ToRA | 13B | `72.7` | `43.0` |
> > > |  |  |  |  |
> > > | WizardMath | 70B | **81.6** | 22.7 |
> > > | WizardMath (ours) | 70B | 80.8 | **24.1** |
> > > | ToRA | 70B | **`84.3`** | 49.7 |
> > > | ToRA-Code | 34B | 80.7 | **`50.8`** |
> > >
> > > As shown in the table above:
> > >
> > > - (1) Our GSM8k evaluation scores align closely with those of WizardMath, differing only by 0.1 to 0.8. The marginally higher scores obtained by WizardMath might be due to its aggressive rounding to the nearest integer, resulting in slightly more false positives. Conversely, our MATH scores are slightly higher (0.5 to 1.4) than WizardMath's, attributable to our use of sympy for result parsing, which leads to more precise evaluation.
> > > - (2) Comparing these results with ToRA, it is evident that, regardless of the grading method used, the ToRA model series has a significant advantage over WizardMath.
> > >
> > > We hope that these supplemental results and responses adequately address your concerns. We welcome any further questions or requests for clarification before the discussion deadline of Nov 22nd.

---

> > > > ### Author Response · Authors · 2023-11-22
> > > >
> > > > Dear Reviewer U6Sz, We hope that our revised paper and the above results address your concerns. If there are any unresolved issues, please let us know. We are more than happy to answer any further questions you may have.

---

### Official Review · Reviewer_V1TN · 2023-10-30

**Soundness:** 3 good
**Presentation:** 3 good
**Contribution:** 3 good
**Rating:** 8
**Confidence:** 3

**Summary:**

This paper introduces TORA (Tool-integrated Reasoning Agents), which seamlessly integrates natural language reasoning with external tools to solve complex mathematical problems. By combining language models' analytical capabilities with computational efficiency tools, TORA significantly outperforms open-source models on 10 mathematical reasoning datasets.

**Strengths:**

1.This paper proposes a two-stage training framework that utilizes training data alternating between natural language and code language to enhance the reasoning ability of language models in mathematical reasoning tasks. The experimental results demonstrate the significant improvement of this approach across 10 datasets.

2.The paper is generally well-written and the figures and tables presented are clear and easy to understand.

**Weaknesses:**

1.From Figure 5, it can be observed that the performance of the model does not significantly decrease when output space shaping is removed. More experiments are needed to demonstrate whether the performance improvement in this stage is due to this training strategy rather than additional data and more training epochs.

2.Regarding the TORA-corpus proposed in this paper, more detailed information is needed regarding the data construction process, quality evaluation, and dataset statistics.

**Questions:**

1.It appears that the method proposed in this paper shows more significant improvements on smaller models, as the performance of the 7B, 13B, and 70B models shown in Figure 1 does not appear to differ significantly. How to explain this phenomenon?

2.The alternating reasoning approach between natural language and tool usage proposed in this paper is fundamentally similar to the plan paradigm of Thought, Action, and Observation alternation in REACT [1]. Do you think this paradigm will become the dominant paradigm for agents to solve complex reasoning problems?
[1] ReAct: Synergizing Reasoning and Acting in Language Models, ICLR 2023

---

> ### Author Response · Authors · 2023-11-16
> **Author Response to Official Review by Reviewer V1TN (1/2)**
>
> Dear reviewer V2TN,
>
> Thank you for your comprehensive and meticulous review. We appreciate your recognition of the excellent performance of ToRA, as well as the approval of our presentation.
>
> ### **On Output Space Shaping Results in Table 5**
>
> > From Figure 5, it can be observed that the performance of the model does not significantly decrease when output space shaping is removed. More experiments are needed to demonstrate whether the performance improvement in this stage is due to this training strategy rather than additional data and more training epochs.
> >
>
> We appreciate your keen observation regarding Figure 5. There was a mistake in Figure 5 in the paper where the “sampling” strategy was Inappropriately referred to as “shaping”, and we will correct this in future updates. Here’s the clarification:
>
> 1. Output Space Shaping, in fact, brought a significant improvement. As shown in Table 3, this method improved the performance of the 7B, 13B, and 34B models by 4.4%, 3.5%, and 3.4% respectively. For instance, ToRA-34B increased from 47.4% to 50.8%. Furthermore, the improvement was particularly noticeable in subtopics like Precalculus, Geometry, and Algebra, with an increase of up to 5.9% ~ 6.3%.
> 2. Output space shaping includes two strategies: sampling and correction. In Figure 5, we conducted three experiments for comparison: pure imitation learning, using only the sampling strategy for shaping, and a combination of sampling + correction strategies for shaping. It's worth noting that all models were trained for the same number of epochs and both shaping experiments used the **same amount of training data**. The results showed that shaping using only sampling led to a 2.7% average improvement, while combining sampling + correction strategies led to an average improvement of 3.4% ~ 4.0% on two datasets under the same data amount. These results demonstrate the effectiveness of the sampling and correction strategies in our proposed output space shaping.
>
> ### **On Including More Detailed Information of ToRA-Corpus**
>
> > Regarding the TORA-corpus proposed in this paper, more detailed information is needed regarding the data construction process, quality evaluation, and dataset statistics.
> >
>
> Thank you for your suggestion! Based on your suggestion, we will add a section in the appendix to provide a more detailed introduction to the data construction process, quality control, and report more data statistical information, beyond Sec. 2.2. Specifically:
>
> - **Data format and quality control**: In our preliminary experiments, we found that the tool-integrated reasoning trajectory format generated by zero-shot prompting was somewhat chaotic. Therefore, we designed a few-shot prompting to control the reasoning format, which effectively improved data quality. On the other hand, we increased the annotation success rate by sampling, ensuring more comprehensive coverage of the training query.
> - **Data filtering process:** For the data constructed, we filtered out paths that produced incorrect answers by matching them with standard answers. To prevent the model from learning incorrect intermediate reasoning processes, we further filtered out data samples with intermediate program execution errors.
> - **Dataset statistics**: We compared the annotation accuracy (i.e., sample coverage) of the training set on GSM8k, MATH, and MATH subtopics of ToRA-Corpus-Greedy (Sec. 2.2) using only the greedy trajectories, and ToRA-Corpus-16k combined with sampled trajectories. Furthermore, we reported the statistical data of ToRA-Corpus-16k, such as the number of samples, average question length, average, minimum, and maximum trajectory length, as shown in the following tables.
>
> *Table 1: Accuracy of ToRA-Corpus-16k on GSM8k and MATH.*
>
> |  | GSM8k | MATH | MATH | MATH | MATH | MATH | MATH | MATH | MATH |
> | --- | --- | --- | --- | --- | --- | --- | --- | --- | --- |
> |  | Overall | Overall | Intermediate Precalculus | Algebra | Geometry | Number Theory | Counting & Probability | Prealgebra | Algebra |
> | ToRA-Corpus-Greedy | 94.4 | 64.3 | 51.0 | 51.5 | 70.0 | 77.4 | 72.2 | 89.8 | 85.1 |
> | ToRA-Corpus-16k | 98.2 | 83.1 | 72.9 | 70.0 | 58.9 | 91.6 | 81.7 | 95.5 | 96.3 |
>
> *Table 2: Statistics of ToRA-Corpus-16k*
>
> |  | GSM8k | MATH | Total |
> | --- | --- | --- | --- |
> | # Train Samples | 7,657 | 7,881 | 15,538 |
> | Avg Question Length | 236 | 189 | 211 |
> | Avg Trajectory Length | 678 | 704 | 691 |
> | Min Trajectory Length | 218 | 119 | 119 |
> | Max Trajectory Length | 1,713 | 2,486 | 2,486 |

---

> ### Author Response · Authors · 2023-11-16
> **Author Response to Official Review by Reviewer V1TN (2/2)**
>
> ### ****On More Significant Improvements on Smaller Models****
>
> > It appears that the method proposed in this paper shows more significant improvements on smaller models, as the performance of the 7B, 13B, and 70B models shown in Figure 1 does not appear to differ significantly. How to explain this phenomenon?
> >
>
> Thank you for your detailed observation! According to Table 2, ToRA-Code's performance on GSM8k increased from 72.6 to 75.8 and 80.7 as the model size increased from 7B to 13B and 34B, respectively. In contrast, the performance on the MATH dataset saturated as the model size increased, with performances of 44.6, 48.1, and 50.8, respectively.
>
> To further understand the bottleneck encountered by ToRA on the MATH dataset at around 50%, we collected the accuracy of different models on MATH problems of different difficulty levels, as shown in the following table:
>
> |  | Level 1 | Level 2 | Level 3 | Level 4 | Level 5 |
> | --- | --- | --- | --- | --- | --- |
> | # Test Samples | 437 | 894 | 1,131 | 1,214 | 1,324 |
> | Avg Question Length | 123.8 | 150.9 | 169.1 | 203.0 | 248.4 |
> | ToRA-Code-7B | 77.3 | 62.3 | 50.0 | 40.9 | 22.3 |
> | ToRA-Code-13B | 78.5 | 64.2 | 54.3 | 45.3 | 26.4 |
> | ToRA-Code-34B | 82.4 | 66.9 | 59.5 | 46.8 | 27.3 |
> | $\Delta$ 13B $\rightarrow$ 34B | +3.9 | +2.7 | +5.2 | +1.5 | +0.9 |
> | GPT-4 PAL | 80.1 | 65.4 | 58.4 | 45.8 | 30.0 |
> | GPT-4 Tool-integrated Reasoning | 89.5 | 77.7 | 71.0 | 55.6 | 39.0 |
> | Training query coverage | 97.7 | 91.6 | 86.5 | 81.3 | 68.0 |
>
> From the table, it is clear that the saturation of performance with increasing model size is primarily evident in difficult problems, i.e., levels 4 and 5. This phenomenon can be attributed to the following reasons:
>
> 1. Firstly, ToRA greatly improved the learning efficiency of smaller models, and raised the starting point of the scaling curve. As shown in Fig. 4, the format of Tool-integrated reasoning is more conducive to model learning under the same amount of training data, allowing smaller models to reach a very high level.
> 2. Secondly, the performance of the largest ToRA model is already close to that of the GPT-4 used for data annotation, and the end point of the scaling curve is mainly limited by the bottleneck of data annotation. The data constructed using GPT-4 has a relatively low coverage of difficult samples (81.3 and 68.0% for Level 4 and Level 5 samples, respectively). Increasing the size of the model increases the model's capacity, but the bottleneck is likely to be the quality and quantity of annotated samples.
>
> This could encourage further research to focus on challenging math problems and more effective data engineering.
>
> ### **On Discussing Alternating Reasoning Paradigm for Agent Reasoning**
>
> > The alternating reasoning approach between natural language and tool usage proposed in this paper is fundamentally similar to the plan paradigm of Thought, Action, and Observation alternation in REACT [1]. Do you think this paradigm will become the dominant paradigm for agents to solve complex reasoning problems?
> >
>
> Thank you for your question! We agree with your view. Since language is the interface for LLMs to interact with tools and environments, the solving process for most LLM-Agent related tasks can be seen as interleaved rationale and interaction trajectory. Therefore, we believe alternating reasoning is a fundamental task completion paradigm based on the language interface, and we look forward to future work further promoting this method to solve more complex reasoning tasks with LLM!

---

> > ### Comment · Reviewer_V1TN · 2023-11-22
> >
> > Thanks for your reply, I have no further questions. I will remain my score.

---

### Official Review · Reviewer_mvjp · 2023-10-30

**Soundness:** 4 excellent
**Presentation:** 4 excellent
**Contribution:** 4 excellent
**Rating:** 8
**Confidence:** 5

**Summary:**

The authors present a framework for improving mathematical reasoning by combining natural language descriptions with program synthesis and execution. A hybrid tool-integrated reasoning context is used to sample candidate reasoning trajectories for the GSM8K and MATH datasets. Candidate trajectories are then verified and those that are successful are added to the ToRA corpus. This corpus is used to fine-tune intermediate models. Such models are then employed as in the initial setting to sample candidate trajectories with feedback from teacher correction to aid the completion of partial trajectories. This final set of valid trajectories is used for further fine-tuning and finally producing the ToRA fleet of models, extended from the LLaMA-2 and CodeLLaMA base families. The resulting models show considerable performance boosts on a range of diverse mathematical reasoning datasets.

**Strengths:**

The idea is clear, well-constructed, and well-explained. The figures are excellent and the algorithm is clearly laid out. The resulting models show considerable performance increases under a range of evaluation settings confirming the efficacy of the strategy.

**Weaknesses:**

While the authors have presented what worked well, there is a considerable amount to be gleaned from the failure modes. The authors loosely allude to failure cases including geometric problems and program timeouts, and provide single examples in the appendix, but there are surely more interesting patterns. It would be wonderful if the authors could provide more specific examples and comment on more systematic classes of errors beyond these simple categorizations. For example, are there certain patterns in the natural language specification of the original problem or rationale construction that fail to formalize well as programs? Were the patterns informative in terms of which problems were amenable to imitation learning vs which required output space shaping in order to produce initial valid reasoning trajectories?

**Questions:**

As noted above, additional discussion of failure modes would be beneficial.

---

> ### Author Response · Authors · 2023-11-16
> **Author Response to Official Review by Reviewer mvjp (1/2)**
>
> Dear Reviewer mvjp,
>
> Thank you for your thoughtful review and the recognition of our work's clarity and effectiveness. We are pleased to provide more in-depth insights into the failure modes and the impact of output space shaping, as per your suggestions.
>
> ### **Adding Comprehensive Failure Mode Analysis**
>
> > While the authors have presented what worked well, there is a considerable amount to be gleaned from the failure modes. The authors loosely allude to failure cases including geometric problems and program timeouts, and provide single examples in the appendix, but there are surely more interesting patterns. It would be wonderful if the authors could provide more specific examples and comment on more systematic classes of errors beyond these simple categorizations. Are there certain patterns in the natural language specification of the original problem or rationale construction that fail to formalize well as programs?
>
> We appreciate your suggestion for a more comprehensive analysis of failure modes. To this end, we manually annotated 100 randomly selected trajectories from the MATH test set, identifying and categorizing their failure modes. Here are the results:
>
> | **Error Type** | **Definition** | **Percentage** |
> | --- | --- | --- |
> | Reasoning Error | Mistakes due to incorrect reasoning steps or missing conditions | 38% |
> | Hallucination | Fabrication of numbers or answers | 5% |
> | Diagram Understanding Error | Misinterpretation of the input diagram | 21% |
> | Inappropriate Tool Usage | Incorrect use of external tools, especially when the problem can't be solved directly with libraries | 10% |
> | Syntax Error | Persistent syntax errors despite multiple correction attempts | 9% |
> | Runtime Error | Errors during program execution, unresolved by retrying | 9% |
> | Rationale-only Error | Cannot be formalized into a program, and the subsequent rationale is also erroneous. | 3% |
> | False Negative | Correct answers that don't fully match the ground truth | 5% |
>
> We observed that:
>
> 1. Incorrect reasoning steps constitute the primary source of errors for ToRA on complex math reasoning tasks (38%), with hallucination issues also evident during problem interpretation and answer finalization (5%).
> 2. Misinterpretation of input diagrams is the second largest category of errors (21%), particularly prevalent in Geometry, Precalculus, and Intermediate Algebra. This may be due to the fact that diagrams in the MATH dataset are often specified in text with the Asymptote language [1], which poses challenges to ToRA when understanding diagrams through text alone.
> 3. Issues with tool usage include Inappropriate Tool Usage (10%), Syntax Error (9%), and Runtime Error (9%). These issues often manifest as ToRA being unable to correctly use tools after multiple rounds of modification and attempts. There are certain inputs that fail to formalize well as programs (3%), which require abstract reasoning rather than computation.
> 4. We also found that there are false negatives when using automatic indicators, i.e., correct predictions that are misjudged as wrong, but the proportion is relatively small (5%).
>
> We will include examples for each failure mode in the Appendix of the updated paper.

---

> ### Author Response · Authors · 2023-11-16
> **Author Response to Official Review by Reviewer mvjp (2/2)**
>
> ### **Understanding the Impact of Output Space Shaping in Relation to Question Difficulty**
>
> > Were the patterns informative in terms of which problems were amenable to imitation learning vs which required output space shaping in order to produce initial valid reasoning trajectories?
> >
>
> Your question about the patterns that determine which problems are amenable to imitation learning versus those that require output space shaping is very interesting!
>
> Initially, we compared the results before and after output space shaping for different subtopics of MATH in Table 3 of the paper. We found that shaping has a more pronounced effect on Precalculus, Geometry, and Algebra, with improvements ranging from 5.9% to 6.3%.
>
> In response to your question, we further compared the effects of output space shaping on MATH problems of different difficulty levels (from level 1 to level 5):
>
> |                     | Level 1  | Level 2  | Level 3  | Level 4  | Level 5  |
> |---------------------|----------|----------|----------|----------|----------|
> | # Test Samples      | 437      | 894      | 1131     | 1214     | 1324     |
> | Avg Question Length | 123.8    | 150.9    | 169.1    | 203.0    | 248.4    |
> | Avg Answer Length   | 503.1    | 655.8    | 751.2    | 881.6    | 1083.8   |
> | ToRA-Code-7B        | 74.1     | 57.5     | 46.9     | 35.2     | 19.4     |
> | + Shaping           | 77.3     | 62.3     | 50.0     | 40.9     | 22.3     |
> | $\Delta$            | +3.2     | **+4.8** | +3.1     | **+5.7** | +2.9     |
> | ToRA-Code-13B       | 78.7     | 63.4     | 48.7     | 39.6     | 21.0     |
> | + Shaping           | 78.5     | 64.2     | 54.3     | 45.3     | 26.4     |
> | $\Delta$            | -0.2     | +0.8     | **+5.6** | **+5.7** | **+5.4** |
> | ToRA-Code-34B       | 79.6     | 65.8     | 54.4     | 43.6     | 24.4     |
> | + Shaping           | **82.4** | **66.9** | **59.5** | **46.8** | 27.3     |
> | $\Delta$            | +2.8     | +1.1     | **+5.1** | **+3.2** | +2.9     |
> | GPT-4 PAL           | 80.1     | 65.4     | 58.4     | 45.8     | **30.0** |
>
> Our findings indicate:
>
> - Across these different difficulty levels, output space shaping generally brings a significant improvement of 4.0% on average across different model sizes.
> - Comparing the growth rates for different difficulty levels, we find that output space shaping brings significant improvements for difficult, long problems. For example, with ToRA-Code-13B, shaping does not significantly improve level 1 to level 2 problems, but it brings a substantial improvement of 5.4% to 5.7% for level 3 to level 5 problems.
> - After using shaping, ToRA-34B outperforms GPT-4 PAL on problems from Level 1 to Level 4, but there is still a gap at Level 5 (27.3% vs. 30.0%). These problems are usually longer (average about 248.4 characters), require more reasoning steps (>1,000 characters) to solve, and more often include diagram inputs (about 20%). These findings may guide future work to focus more on solving these more difficult problems.
>
> We will include these analyses in the updated version of the paper. Once again, thank you for your excellent suggestions!
>
> ### References
>
> [1] Hendrycks, Dan, et al. "Measuring mathematical problem solving with the math dataset." arXiv preprint arXiv:2103.03874 (2021).

---

> > ### Comment · Reviewer_mvjp · 2023-11-19
> > **Reviewer Response to Author Comments**
> >
> > Thank you for your reply. These additional findings are indeed useful and informative, highlighting opportunities for future contribution.
> >
> > I recommend acceptance. Great job!

---

### Public Comment · ~Wei_Luo8 · 2023-11-27
**Most solutions consist of one step of natural language reasoning and program-based tool use, seem to be inconsistent with the approach mentioned in the paper.**

Thank you for this excellent research work. I have two questions:

1. The example "Listing 3: Success case for TORA: Self-Correcting Errors with tool feedback" from Appendix is from the MATH training set, with the question id 267, and it is not from the MATH test set. Why this example is considered a success case for TORA? Maybe it is better to give an example from the test set that demonstrates "Self-Correcting Errors with tool feedback."

2. We found that there are very few cases of multi-step or looping scenarios in the actual data.
- Regarding this point, there are three pieces of evidence:
  - The examples provided in Appendix D (list 2, 4, 5, 6; listing 3 is an incorrect case). No data includes two or more steps.
  - The training set examples provided on GitHub (https://github.com/microsoft/ToRA/blob/main/data/tora/examples.jsonl). No data includes two or more steps.
  - We performed inference on the MATH training set using the tora-code-7b-v1.0 model (https://huggingface.co/llm-agents/tora-code-7b-v1.0) based on greedy search. No data includes two or more steps.
- If the majority of the data exhibits the aforementioned patterns, with only one step of natural language reasoning and program-based tool use, then the method described in the paper and Figure 2 both mention the need for looping iterations of natural language reasoning and program-based tool use. Is there a gap in this regard?

---

### Meta-Review · Area_Chair_8kAE · 2023-12-10

**Metareview:**

The work integrates LLMs with external tools to address complex mathematical problems, by learning from interactive tool-use trajectories. Overall the reviewers think the problem is well motivated, the method is well designed, and the experimental results are significant. One main concern is that the main framework, including imitation learning and alternating reasoning between LLMs and external tools, has been widely used in previous and several concurrent works. AC agrees although the framework is not new, the solution is well designed and effective, and thus recommends acceptance as poster. The camera version should include the discussions on similarities and differences to related (including concurrent) works, and open-source as it compared mostly to open-sourced models.

**Justification For Why Not Higher Score:**

While the solution is well designed and effective, the framework has been widely used in previous and concurrent works.

**Justification For Why Not Lower Score:**

The problem is important, the method is well designed, and the performance is significant.

---

### Decision · Program_Chairs · 2024-01-16

Accept (poster)